# SE-Diff: Simulator and Experience Enhanced Diffusion Model for Comprehensive ECG Generation

**Xiaoda Wang**[*], **Kaiqiao Han**[†], **Yuhao Xu**[*], **Xiao Luo**[‡], **Yizhou Sun**[†], **Wei Wang**[†], **Carl Yang**[*]
[*]Emory University  [†]University of California, Los Angeles  [‡]University of Wisconsin–Madison
{xiaoda.wang,yxu81,j.carlyang}@emory.edu   xiao.luo@wisc.edu
{kqhan,yzsun,weiwang}@cs.ucla.edu

## Abstract

Cardiovascular disease (CVD) is a leading cause of mortality worldwide. Electrocardiograms (ECGs) are the most widely used non-invasive tool for cardiac assessment, yet large, well-annotated ECG corpora are scarce due to cost, privacy, and workflow constraints. Generating ECGs can be beneficial for the mechanistic understanding of cardiac electrical activity, enable the construction of large, heterogeneous, and unbiased datasets, and facilitate privacy-preserving data sharing. Generating realistic ECG signals from clinical context is important yet underexplored. Recent work has leveraged diffusion models for text-to-ECG generation, but two challenges remain: (i) existing methods often overlook the physiological simulator knowledge of cardiac activity; and (ii) they ignore broader, experience-based clinical knowledge grounded in real-world practice. To address these gaps, we propose **SE-Diff**, a novel physiological simulator and experience enhanced diffusion model for comprehensive ECG generation. SE-Diff integrates a lightweight ordinary differential equation (ODE)-based ECG simulator into the diffusion process via a beat decoder and simulator-consistent constraints, injecting mechanistic priors that promote physiologically plausible waveforms. In parallel, we design an LLM-powered experience retrieval-augmented strategy to inject clinical knowledge, providing more guidance for ECG generation. Extensive experiments on real-world ECG datasets demonstrate that SE-Diff improves both signal fidelity and text–ECG semantic alignment over baselines, proving its superiority for text-to-ECG generation. We further show that the simulator-based and experience-based knowledge also benefit downstream ECG classification.

## 1 Introduction

Cardiovascular disease (CVD) remains a leading cause of global mortality and morbidity (Roth et al., 2020). In clinical workflows, the 12-lead electrocardiogram (ECG), the standard setup using 10 electrodes to derive 12 voltage traces, is ubiquitous, non-invasive, and low-cost for screening, triage, and longitudinal monitoring (Kligfield et al., 2007; Xu et al., 2025a;b). While machine learning (ML) has advanced ECG interpretation, progress is constrained by limited access to large, well-annotated corpora, stringent privacy considerations around sharing protected health information, and the expense of expert labeling (Johnson et al., 2023; Goldberger et al., 2000; Wang et al., 2026; 2025b). ECG generation provides a principled way to mitigate these barriers by expanding training data, enabling controlled curation, and decoupling model development from directly identifiable records (Zanchi et al., 2025). Concurrently, denoising diffusion probabilistic models (DDPMs) and score-based methods have established strong fidelity and coverage across modalities (Ho et al., 2020; Song et al., 2021), motivating their adaptation to time series and, specifically, text-conditioned ECG generation (Lai et al., 2025a).

---

Corresponding author: Carl Yang (j.carlyang@emory.edu). Code is available at https://github.com/ignite-abd/SE-Diff.

Despite these advances, there are still two gaps limit the practical adoption of text-to-ECG generation. *(i) Missing physiological simulator knowledge.* Most diffusion models for ECG learn morphology and timing purely from data, with minimal incorporation of known cardiac physiological dynamics. Decades of physiological modeling have produced compact ordinary differential equation (ODE) simulators that yield realistic P–QRS–T morphologies and heart rate variability under controllable parameters (McSharry et al., 2003; Malik, 1996). Yet these simulators are rarely integrated as priors or constraints during diffusion training, leaving a disconnect between statistical generation and mechanistic plausibility. *(ii) Under-use of experience-based knowledge at scale.* Prior text-to-ECG works often condition on narrow patient metadata, but do not leverage broader *experience knowledge*—case-based regularities distributed across large electronic health record (EHR) corpora. Retrieval-augmented generation (RAG) offers a principled means to inject such non-parametric knowledge to generators (Lewis et al., 2020), including via lexical retrieval schemes, yet its potential for conditioning medical time-series generation remains underexplored.

To address these challenges, we introduce SE-Diff, a conditional latent-diffusion framework that synthesizes comprehensive ECG waveforms from natural-language clinical descriptions. SE-Diff couples a lightweight ODE-based ECG simulator to the denoising dynamics through a beat decoder—reconstructing a QRS-aligned single cycle from the latent code—and simulator-consistent spectral and rate constraints, thereby injecting mechanistic priors that steer generation toward physiologically plausible signals. In parallel, an LLM-powered retrieval pipeline identifies clinically similar patients from EHRs, retrieves their ECG diagnoses and measurements, and distills them into a concise, physiologically grounded description that is fused with available metadata for conditioning. In summary, our main contributions are as follows:

❶ **Problem Identification.** We identify the problem of generating realistic 10-second, 12-lead ECG waveforms directly from natural-language clinical descriptions. We propose SE-Diff, which can incorporate various patient metadata (age, sex, heart rate, rhythm/conduction) as soft clinical constraints to steer morphology toward clinically meaningful generation.

❷ **Simulator–Informed Diffusion.** SE-Diff is the first to integrate a lightweight ODE-based ECG simulator with latent diffusion. We introduce a beat decoder that reconstructs a single-cycle beat from the latent representation, injecting simulator-consistent mechanistic priors that guide the denoising process toward physiologically plausible waveforms.

❸ **Experience Retrieval–Augmented Conditioning.** We design an LLM-powered retrieval pipeline that identifies clinically similar patients based on EHR data and retrieves ECG diagnoses and measurements. The LLM generates a concise, physiologically grounded description, which is fused with available metadata to form the conditioning context.

❹ **Experimental Validation.** Across real-world ECG datasets, SE-Diff surpasses baselines in both signal fidelity and text–ECG semantic alignment. Ablations quantify the contribution of simulator-based and experience-based knowledge conditioning. We further show that SE-Diff improves downstream ECG classification when used for augmentation.

## 2 PRELIMINARIES

### 2.1 DENOISING DIFFUSION PROBABILISTIC MODELS

Denoising Diffusion Probabilistic Models (DDPMs) (Sohl-Dickstein et al., 2015; Ho et al., 2020) define a fixed forward Markov noising process that maps a clean sample $x_0 \sim q(x_0)$ to Gaussian noise over $T$ steps, and a parametric reverse process that approximately inverts it. With variance schedule $\{\beta_t\}_{t=1}^T \subset (0,1)$, set $\alpha_t = 1 - \beta_t$ and $\bar{\alpha}_t = \prod_{s=1}^t \alpha_s$. The forward chain is

$$q(x_{1:T} \mid x_0) = \prod_{t=1}^T q(x_t \mid x_{t-1}), \tag{1}$$

where $q(x_t \mid x_{t-1}) := \mathcal{N}(\sqrt{\alpha_t}\, x_{t-1},\, \beta_t\, \mathbf{I})$. This implies the closed form $x_t = \sqrt{\bar{\alpha}_t}\, x_0 + \sqrt{1 - \bar{\alpha}_t}\, \epsilon_t$, with $\epsilon_t \sim \mathcal{N}(0, \mathbf{I})$. The exact reverse-time posterior $q(x_{t-1} \mid x_t)$ is intractable, so DDPMs approximate it with a Gaussian transition $p_\theta(x_{t-1} \mid x_t) := \mathcal{N}(\mu_\theta(x_t, t),\, \Sigma_\theta(x_t, t))$, where a neural network predicts either the forward noise $\epsilon$, the clean signal $x_0$, or the velocity $v$. Under the com-

mon noise-prediction parameterization with $\epsilon_\theta(x_t, t)$, the mean is

$$\mu_\theta(x_t, t) \;=\; \frac{1}{\sqrt{\alpha_t}} \left( x_t - \frac{\beta_t}{\sqrt{1 - \bar{\alpha}_t}} \, \epsilon_\theta(x_t, t) \right), \tag{2}$$

and we set $\Sigma_\theta(x_t, t) \in \{\beta_t \mathbf{I}, \sigma_t^2 \mathbf{I}\}$. Training maximizes a variational lower bound on $\log p_\theta(x_0)$ (Sohl-Dickstein et al., 2015), which in practice reduces to the simple loss (Ho et al., 2020) with optional step-dependent weights $w_t$:

$$\mathcal{L}_{\text{simple}}(\theta) = \mathbb{E}_{t \sim \mathcal{U}\{1...T\}, \, x_0 \sim q, \, \epsilon \sim \mathcal{N}} \Big[ w_t \, \big\| \epsilon - \epsilon_\theta\big(\sqrt{\bar{\alpha}_t}x_0 + \sqrt{1 - \bar{\alpha}_t}\, \epsilon, \, t\big) \big\|_2^2 \Big]. \tag{3}$$

## 2.2 The ECG Physiological Simulator

In a resting heart, the ECG follows the P–QRS–T sequence. To reproduce this morphology, McSharry et al. (2003) proposed a three–ODE "ECG simulator" that generates realistic P–QRS–T waves while allowing control of heart-rate statistics and HRV spectrum (Malik & Camm, 1990). The model evolves a 3D state $(x(t), y(t), z(t))$: $(x, y)$ traverse a unit-radius limit cycle whose angle encodes cardiac phase (one revolution per beat), and $z(t)$ is the ECG voltage given by excursions about this cycle. The governing ODEs are:

$$\frac{dx}{dt} = \alpha(x, y)\, x \, - \, \omega\, y, \qquad \frac{dy}{dt} = \alpha(x, y)\, y \, + \, \omega\, x \,. \tag{4}$$

$$\frac{dz}{dt} = - \sum_{\beta \in \{P,Q,R,S,T\}} a_\beta \, \Delta\theta_\beta(x, y) \, \exp\Big( - \frac{\Delta\theta_\beta(x, y)^2}{2\, b_\beta^2} \Big) \, - \, \big[\, z - z_0(t)\,\big] \,. \tag{5}$$

where $\alpha(x, y) = 1 - \sqrt{x^2 + y^2}$ drives $(x, y)$ toward the unit limit cycle, $\theta(x, y) = \text{atan2}(y, x) \in [-\pi, \pi]$ is the phase, and $\Delta\theta_\beta(x, y) = (\theta(x, y) - \theta_\beta) \bmod 2\pi$ is the phase offset to landmark $\beta \in \mathcal{B}$ with $\mathcal{B} = \{P, Q, R, S, T\}$. The parameter $\omega$ controls angular velocity (thus average heart rate), and $z_0(t)$ is a slow baseline (e.g., respiratory wander modeled as $z_0(t) = A \sin(2\pi f_{\text{resp}} t)$ with small amplitude $A$ (Sörnmo & Laguna, 2005)). All morphology parameters are collected as

$$\eta \;=\; \big\{ \theta_P, \theta_Q, \theta_R, \theta_S, \theta_T, \;\; a_P, a_Q, a_R, a_S, a_T, \;\; b_P, b_Q, b_R, b_S, b_T \big\}, \tag{6}$$

where these parameters—phase landmarks $\theta_\beta$, amplitude coefficients $a_\beta$, and width coefficients $b_\beta$ for each $\beta \in \{P, Q, R, S, T\}$—govern the ECG morphology. When the $(x, y)$ state passes the phase $\theta_\beta$, the Gaussian-shaped term $a_\beta \, \Delta\theta_\beta \exp\big( - \Delta\theta_\beta^2/(2b_\beta^2) \big)$ in 5 transiently perturbs $z$ away from baseline, producing the corresponding P/QRS/T deflection. The sign of $a_\beta$ sets polarity (upward for $a_\beta > 0$, downward for $a_\beta < 0$); $|a_\beta|$ controls peak amplitude; and $b_\beta$ sets the temporal spread (wave duration). The restoring term $-[z - z_0(t)]$ then pulls the signal back toward baseline. Unless otherwise specified, we adopt the parameter values recommended by McSharry et al. (2003).

**The Euler Method.** To simulate the synthetic ECG $z(t)$, we numerically solve the ODE system with a fixed-step explicit Euler method (the first-order Runge–Kutta scheme) (Butcher & Butcher, 1987; Süli & Mayers, 2003). We choose the step size $\Delta t = 1/f_s$ to match the desired sampling frequency (e.g., $f_s = 500$ Hz). Using the finite-difference approximation (Milne-Thomson, 2000):

$$\frac{du}{dt}(t) \;\approx\; \frac{u(t + \Delta t) - u(t)}{\Delta t} \,, \tag{7}$$

which leads to the update rule $u(t + \Delta t) = u(t) + v(t)\, \Delta t$, for an ODE of the form $du/dt = v(t)$. Starting from initial conditions $(x_0, y_0, z_0)$, we iterate this update for each time step. At the $\ell$-th step (time $t_\ell = \ell \, \Delta t$), let $v_\ell = \big( f_x(x_\ell, y_\ell; \eta), \;\; f_y(x_\ell, y_\ell; \eta), \;\; f_z(x_\ell, y_\ell, z_\ell, t_\ell; \eta) \big)$ denote the right-hand side of Equation 4 and 5. The state is then advanced as:

$$x_{\ell+1} = x_\ell + f_x(x_\ell, y_\ell; \eta)\, \Delta t \,, \tag{8}$$

$$y_{\ell+1} = y_\ell + f_y(x_\ell, y_\ell; \eta)\, \Delta t \,, \tag{9}$$

$$z_{\ell+1} = z_\ell + f_z(x_\ell, y_\ell, z_\ell, t_\ell; \eta)\, \Delta t \,, \tag{10}$$

and this process is repeated for $\ell = 0, 1, 2, \ldots$ up to the desired number of samples $L$. In other words, each iteration uses the derivatives $f_x, f_y, f_z$ at the current state to step the solution forward by $\Delta t$. This simple explicit scheme is computationally efficient and sufficient for our purposes, though higher-order integration methods could be used for greater accuracy if needed.

# 3 METHOD: SE-DIFF

We present SE-Diff, a conditional latent-diffusion framework that synthesizes 10s, 12-lead ECGs from clinical text. Diffusion operates in the VAE latent space (Sec. 2.1). To make physiology-aware supervision tractable, we attach a lightweight *Beat Decoder* that predicts a single QRS-aligned cardiac cycle from the latent; its output drives simulator-informed regularizers derived from the ECG physiology model in Sec. 2.2. To strengthen conditioning, SE-Diff also incorporates experience knowledge retrieved based on EHRs (Sec. 3.4). In inference, we sample in latent space and decode with the full VAE.

**Problem Formulation.** Each ECG record is a multivariate sequence $\mathbf{x} \in \mathbb{R}^{12 \times L}$ representing a 10 s, 12-lead waveform sampled at $f_s$. Our goal is to learn a conditional generator $p(\mathbf{x} \mid c; \phi, \vartheta, \theta)$ that uses $c$ throughout denoising to produce physiologically plausible ECGs. The conditioning is $c = (t, m, r)$, comprising original diagnoses $t$, basic metadata $m$ (age, sex, optionally heart rate), and retrieve-augmented report $r$. Concretely, we first train a VAE $(E_\phi, D_\theta)$ together with a *Beat Decoder* $D_\psi^{\text{beat}}$; the encoder maps a full recording to a latent sequence $z_0 = E_\phi(\mathbf{x}) \in \mathbb{R}^{d \times T}$, where $T = L/S$ and $S$ is the VAE temporal stride, and $D_\psi^{\text{beat}}$ maps $z_0$ to a single-cycle prediction $h \in \mathbb{R}^{12 \times L_c}$. We then freeze $E_\phi$, $D_\theta$, and $D_\psi^{\text{beat}}$ and train a DDPM in latent space using a U-Net denoiser $\epsilon_\vartheta(z_t, t, c)$ with cross-attention to $c$. During diffusion training, simulator-guided penalties (Sec. 3.3) are applied to the Beat Decoder output $D_\psi^{\text{beat}}(z_0)$, while experience–knowledge features augment the text pathway (Sec. 3.4). At test time, we run the reverse process to obtain $\hat{z}_0$ and decode $\hat{\mathbf{x}} = D_\theta(\hat{z}_0)$ (Sec. 3.5).

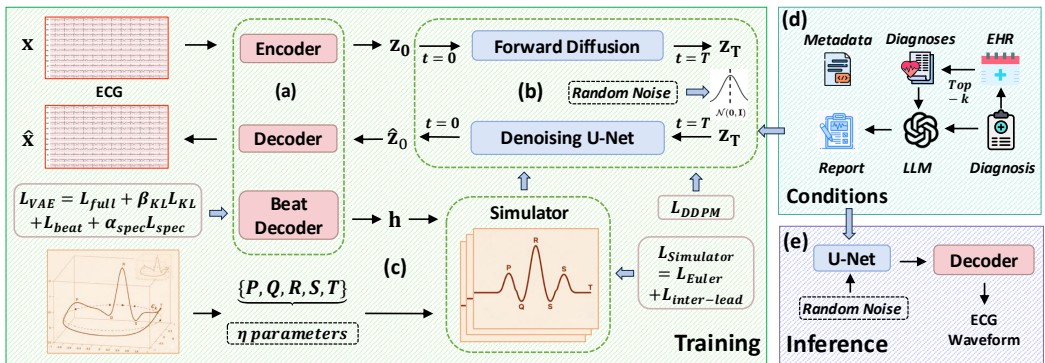

Figure 1: **Overview Framework of SE-Diff.** (a) *Variational Autoencoder*: encoder–decoder with a lightweight beat decoder for a single QRS-aligned cycle. (b) *Conditional latent diffusion*: U-Net denoiser with cross-attention to text, metadata, and retrieved report. (c) *Simulator-informed diffusion*: Euler and inter-lead constraints on the beat decoder output. (d) *Experience retrieval–augmented Conditioning*: tri-view EHR similarity with LLM distillation into a concise report. (e) *Inference*: reverse diffusion and decoding to a 10-second, 12-lead ECG.

## 3.1 VARIATIONAL AUTOENCODER

We learn a latent representation for 12-lead ECGs with a variational autoencoder (VAE). Given a full recording $\mathbf{x} \in \mathbb{R}^{12 \times L}$, the encoder $E_\phi$ parameterizes a diagonal Gaussian posterior

$$q_\phi(z \mid \mathbf{x}) = \mathcal{N}\big(z; \ \mu_\phi(\mathbf{x}), \ \text{diag}(\sigma_\phi^2(\mathbf{x}))\big), \qquad z_0 = \mu_\phi(\mathbf{x}) + \sigma_\phi(\mathbf{x}) \odot \epsilon, \ \epsilon \sim \mathcal{N}(0, \mathbf{I}), \quad (11)$$

where $z_0 \in \mathbb{R}^{d \times T}$ with $T = L/S$ and $S$ the VAE temporal stride. The decoder $D_\theta$ reconstructs the signal $\hat{\mathbf{x}} = D_\theta(z_0) \in \mathbb{R}^{12 \times L}$. To expose morphology at the beat scale, we attach a lightweight Beat Decoder: $D_\psi^{\text{beat}} : \mathbb{R}^{d \times T} \to \mathbb{R}^{12 \times L_c}$ to get the single cycle signal: $h = D_\psi^{\text{beat}}(z_0)$.

**Training.** Let $r_0$ denote the first R-peak index at sampling rate $f_s$ (Golany et al., 2020b); define $\mathcal{C}(\mathbf{x}) = \mathbf{x}[:, r_0 - 0.2 f_s : r_0 + 0.4 f_s]$ and $L_c = 0.2 f_s + 0.4 f_s$. We train the encoder, decoder and Beat Decoder with length-normalized mean-squared errors (MSE) and a single KL term:

$$\mathcal{L}_{\text{full}} = \frac{1}{12L} \left\| \mathbf{x} - D_\theta(E_\phi(\mathbf{x})) \right\|_F^2, \qquad \mathcal{L}_{\text{KL}} = \text{KL}(q_\phi(z \mid \mathbf{x}) \| \mathcal{N}(0, \mathbf{I})), \quad (12)$$

$$\mathcal{L}_{\text{beat}} = \frac{1}{12L_c} \left\| \mathcal{C}(\mathbf{x}) - D_\psi^{\text{beat}}(E_\phi(\mathbf{x})) \right\|_F^2 . \tag{13}$$

The Beat Decoder's single beat should also reflect the statistics of all beats in the $10\,\text{s}$ window. Rather than tiling $h$ to length $L$, we detect all R-peaks within the window, $\{r_j\}_{j=1}^J$ (with $J$ determined by the number of detected beats), and extract per-beat crops of identical length $L_c$: $\mathcal{C}_j(\mathbf{x}) = \mathbf{x}[:, r_j - 0.2\,f_s : r_j + 0.4\,f_s] \in \mathbb{R}^{12 \times L_c}$, $j = 1, \ldots, J$. Let the Beat Decoder output be $h = D_\psi^{\text{beat}}(z_0) \in \mathbb{R}^{12 \times L_c}$. For each lead $\ell$ and each cycle $j$, we remove the mean and compute the one-sided log-magnitude spectrum (real FFT of length $L_c$) up to $f_{\max}$:

$$S_\ell^{(j)}[k] = \log\big(\varepsilon + \big|\text{rFFT}(\mathcal{C}_j(\mathbf{x})_\ell - \overline{\mathcal{C}_j(\mathbf{x})_\ell})\big|[k]\big), \qquad \hat{S}_\ell[k] = \log\big(\varepsilon + \big|\text{rFFT}(h_\ell - \bar{h}_\ell)\big|[k]\big], \tag{14}$$

with $\varepsilon > 0$ small and frequencies $f_k = \frac{k}{L_c} f_s$. The spectral loss averages the (optionally band-weighted) squared discrepancy between the Beat Decoder's spectrum and the spectrum of each observed cycle:

$$\mathcal{L}_{\text{spec}} = \frac{1}{12JK} \sum_{\ell=1}^{12} \sum_{j=1}^{J} \sum_{k:\, f_k \leq f_{\max}} w(f_k) \left( \hat{S}_\ell[k] - S_\ell^{(j)}[k] \right)^2, \tag{15}$$

where $K = |\{k : f_k \leq f_{\max}\}|$ and $w(f)$ can emphasize physiologically salient bands (e.g., higher weights on $0.5$–$3\,\text{Hz}$ for heart rate). We jointly optimize the encoder, the full decoder, and the beat decoder with:

$$\mathcal{L}_{\text{VAE}} = \mathcal{L}_{\text{full}} + \beta_{\text{KL}}\,\mathcal{L}_{\text{KL}} + \mathcal{L}_{\text{beat}} + \alpha_{\text{spec}}\,\mathcal{L}_{\text{spec}}, \tag{16}$$

where $\alpha_{\text{spec}} > 0$. Length normalization makes $\mathcal{L}_{\text{full}}$ and $\mathcal{L}_{\text{beat}}$ commensurate; $\mathcal{L}_{\text{spec}}$ incorporates full-window frequency statistics into the single-cycle prediction.

## 3.2 CONDITIONAL LATENT DIFFUSION

We train the diffusion in the VAE latent space. Given a latent sequence $z_0 = E_\phi(\mathbf{x}) \in \mathbb{R}^{d \times T}$, the forward process follows Sec. 2.1 with $x_t \mapsto z_t$: $z_t = \sqrt{\bar{\alpha}_t}\,z_0 + \sqrt{1 - \bar{\alpha}_t}\,\epsilon$, and $\epsilon \sim \mathcal{N}(0, \mathbf{I})$. We train a conditional denoiser $\epsilon_\vartheta(z_t, t, c)$ with the standard objective

$$\mathcal{L}_{\text{DDPM}} = \mathbb{E}_{t,\,z_0,\,\epsilon}\left[ \left\| \epsilon - \epsilon_\vartheta(z_t, t, c) \right\|_2^2 \right]. \tag{17}$$

The denoiser is a 1D U-Net (Ronneberger et al., 2015) that treats the latent as a sequence $z_t \in \mathbb{R}^{d \times T}$ (channels $d$, length $T$). Conditioning enters via cross-attention to a context representation $C$ built from $c = (t, m, r)$ (Rombach et al., 2022; Vaswani et al., 2017); a final $1 \times 1$ convolution maps features to $\epsilon_\vartheta$. Timestep and context embeddings modulate intermediate features through FiLM-style affine transformations (Perez et al., 2018). At sampling, we use standard DDPM transitions with classifier-free guidance (Ho & Salimans, 2022) and common improvements such as cosine schedules and optional learned variances (Nichol & Dhariwal, 2021); further architectural and training details are provided in the appendix.

## 3.3 SIMULATOR-INFORMED DIFFUSION

We estimate class-specific parameters $\eta_{\text{class}}$ offline by fitting the simulator to representative real beats using an explicit Euler integrator, together with lightweight stabilizers and morphology priors that improve convergence and preserve physiological plausibility (details in Appendix B). For each training sample, we obtain a single-cycle waveform $h = D_\psi^{\text{beat}}(z_0) \in \mathbb{R}^{12 \times L_c}$ from the Beat Decoder (Sec. 3.1). This beat is used to enforce mechanistic plausibility via an ODE-based ECG simulator (Sec. 2.2). The simulator's morphology parameters $\eta = \{\theta_\beta, a_\beta, b_\beta\}_{\beta \in \{P,Q,R,S,T\}}$ enter the right-hand side $f_z(\cdot\,; \eta)$ of Eq. 5, which defines $f_z$. During diffusion training, the simulator provides two complementary regularizers:

**Simulator-guided Euler Loss.** Given the single-cycle waveform $h \in \mathbb{R}^{12 \times L_c}$, we integrate the simulator with parameters $\eta$ and fixed initials $(x_0, y_0)$ to obtain a reference trajectory $(x_t, y_t)$ at $t = \ell \Delta t$. We penalize per-lead deviations from the ODE and the simulator-guided Euler loss is:

$$\mathcal{L}_{\text{Euler}} = \frac{1}{12(L_c - 1)} \sum_{\text{lead}} \sum_{\ell=1}^{L_c - 1} \left( \frac{h_{\ell+1} - h_\ell}{\Delta t} - f_z(x_\ell, y_\ell, h_\ell, t_\ell; \eta) \right)^2. \tag{18}$$

**Inter-lead dependency constraint.** Realistic 12-lead synthesis requires not only accurate per-lead morphology but also correct physiological interdependencies among leads. We therefore enforce the classical frontal-plane identities implied by the standard ECG configuration (Einthoven's triangle and Goldberger's central terminal), constraining the generated limb and augmented leads to remain mutually consistent:

$$
\begin{aligned}
I &= II - III, \quad aVR = -\tfrac{1}{2}(I + II), \quad aVL = \tfrac{1}{2}(I - III), \\
II &= I + III, \quad aVF = \tfrac{1}{2}(II + III), \quad III = II - I.
\end{aligned}
\tag{19}
$$

Let $\mathcal{L}_{\text{frontal}} = \{I, II, III, aVR, aVL, aVF\}$ be the frontal-plane leads (a subset of the 12 leads). For any identity of the form $y = \beta\, p + \gamma\, q$ with $y, p, q \in \mathcal{L}_{\text{frontal}}$, we treat $y$ as the *child* lead and $p, q$ as its *parent* leads. Denote by $h_\ell^L$ the sample at index $\ell$ of lead $L$ from the predicted 12-lead beat $h \in \mathbb{R}^{12 \times L_c}$. We obtain parent simulator states $(x_\ell^p, y_\ell^p)$ and $(x_\ell^q, y_\ell^q)$ by integrating the simulator with class-specific parameters $\eta_p, \eta_q$. Defining $\mathcal{C} = \{(I, II, III, 1, -1),\ (II, I, III, 1, 1),\ (III, II, I, 1, -1),\ (aVR, I, II, -\tfrac{1}{2}, -\tfrac{1}{2}),\ (aVL, I, III, \tfrac{1}{2}, -\tfrac{1}{2}),\ (aVF, II, III, \tfrac{1}{2}, \tfrac{1}{2})\}$ and the loss aggregates the six constraints over time:

$$
\mathcal{L}_{\text{inter-lead}} = \sum_{(y,p,q,\beta,\gamma) \in \mathcal{C}} \sum_{\ell=1}^{L_c-1} \left( \frac{h_{\ell+1}^y - h_\ell^y}{\Delta t} - \beta\, f_z(x_\ell^p, y_\ell^p, h_\ell^p, t_\ell; \eta_p) - \gamma\, f_z(x_\ell^q, y_\ell^q, h_\ell^q, t_\ell; \eta_q) \right)^2.
\tag{20}
$$

Here $y, p, q$ are specific leads (elements of the 12-lead set), and $\mathcal{C}$ enumerates each frontal-plane identity as a tuple $(y, p, q, \beta, \gamma)$. This construction directly matches the child's discrete derivative to the corresponding linear combination of the parents' simulator derivatives, enforcing physiologically grounded inter-lead consistency.

## 3.4 EXPERIENCE RETRIEVAL–AUGMENTED CONDITIONING

We augment text conditioning with clinical experience retrieved from electronic health records (EHR). Specifically, we link MIMIC-IV-ECG (Gow et al., 2023) to MIMIC-IV-CLINICAL (Johnson et al., 2023), build a compact tri-view profile (diagnoses, medications, procedures) (Ou et al., 2025), and retrieve the top-$k$ clinically similar admissions. For an index admission $u$, let $E_u^{\text{Diag}}$, $E_u^{\text{Med}}$, and $E_u^{\text{Proc}}$ denote the sets of diagnosis, medication, and procedure codes, respectively. Given another admission $u'$, we compute set similarities using the Jaccard index $J(A, B)$ for $X \in \{\text{Diag}, \text{Med}, \text{Proc}\}$:

$$
\tau_{\text{X}}(u, u') = J\big(E_u^{\text{X}},\, E_{u'}^{\text{X}}\big).
\tag{21}
$$

These similarities are aggregated with nonnegative weights $\lambda_1, \lambda_2, \lambda_3$ to yield a single similarity:

$$
\tau(u, u') = \lambda_1 \tau_{\text{Diag}}(u, u') + \lambda_2 \tau_{\text{Med}}(u, u') + \lambda_3 \tau_{\text{Proc}}(u, u').
\tag{22}
$$

We retrieve the top-$k$ most similar admissions according to the tri-view similarity over diagnoses, medications, and procedures, and then pass their diagnostic profiles together with $(t, m)$ to the LLM using the prompt shown in Fig. 5 in the Appendix to obtain a concise, physiologically grounded report $r$, where $t$ denotes the original diagnoses and $m$ encodes basic metadata (age, sex, optionally heart rate). Finally, the conditioning input is $c = (t, m, r)$, which conditions the denoiser via cross-attention.

## 3.5 TRAINING AND INFERENCE

**Training objective.** We combine the latent-space diffusion loss with simulator-based regularizers:

$$
\mathcal{L}_{\text{total}} = \mathcal{L}_{\text{DDPM}} + \lambda\, \mathcal{L}_{\text{Euler}} + \gamma\, \mathcal{L}_{\text{inter-lead}}, \qquad \lambda, \gamma > 0.
\tag{23}
$$

We first train the VAE using Eq. 16 and then freeze $E_\phi$, $D_\theta$, and $D_\psi^{\text{beat}}$. During diffusion training, we optimize only the denoiser $\epsilon_\vartheta$; the Beat Decoder appears only through these regularizers—we use $D_\psi^{\text{beat}}$ to produce $h = D_\psi^{\text{beat}}(z_0)$ for $\mathcal{L}_{\text{Euler}}$ and $\mathcal{L}_{\text{inter-lead}}$. All simulator-driven terms are training-only and do not modify the reverse process.

**Inference.** Given conditioning $c$, we draw $z_T \sim \mathcal{N}(0, \mathbf{I})$ and apply the learned reverse diffusion from $t = T$ to 1 with the standard DDPM parameterization (variance schedule $\{\beta_t\}$, $\alpha_t = 1 - \beta_t$,

$\bar{\alpha}_t = \prod_{s=1}^{t} \alpha_s$):

$$\hat{z}_0(z_t, t, c) = \frac{z_t - \sqrt{1 - \bar{\alpha}_t}\, \epsilon_\vartheta(z_t, t, c)}{\sqrt{\bar{\alpha}_t}}, \tag{24}$$

$$\mu_\vartheta(z_t, t, c) = \frac{1}{\sqrt{\alpha_t}} \left( z_t - \frac{\beta_t}{\sqrt{1 - \bar{\alpha}_t}}\, \epsilon_\vartheta(z_t, t, c) \right). \tag{25}$$

We set $\tilde{\beta}_t = \frac{1 - \bar{\alpha}_{t-1}}{1 - \bar{\alpha}_t} \beta_t$ and sample $z_{t-1} = \mu_\vartheta(z_t, t, c) + \sqrt{\tilde{\beta}_t}\, \xi_t$ with $\xi_t \sim \mathcal{N}(0, \mathbf{I})$. After the final step, we decode to the signal domain, $\hat{\mathbf{x}} = D_\theta(\hat{z}_0)$, optionally using classifier-free guidance during sampling.

## 4 EXPERIMENTS

**Dataset and Preprocessing.** We train on MIMIC-IV-ECG (Gow et al., 2023; Johnson et al., 2023), which contains 800,035 de-identified 12-lead, 10 s ECGs sampled at 500 Hz. Heart rate (HR) is taken from metadata when available; otherwise it is re-estimated via QRS detection (WFDB XQRS). Waveforms are encoded by a VAE into $4 \times 128$ latents that serve as inputs to the diffusion model. We use the MIMIC-IV-Clinical (Johnson et al., 2023) to obtain each patient's EHR for experience knowledge conditioning. For external validation, we additionally experiment on the PTB-XL dataset (Wagner et al., 2020), which consists of 12-lead clinical ECGs with standardized diagnostic labels.

**Baselines.** We compare against four strong ECG generation baselines: (i) *DiffuSETS* (Lai et al., 2025a), to our knowledge the only prior method that generates *12-lead, 10 s* ECGs from clinical text; (ii) a GAN-based model (WGAN) originally proposed for arrhythmia classification using cGAN-augmented ECG signals (Adib et al., 2022), which we adapt to synthesize 10 s, 12-lead ECGs under our setting; (iii) SSSD, a diffusion-based conditional ECG generator built on structured state space models (López Alcaraz & Strodthoff, 2023); and (iv) BeatDiff, an ECG beat diffusion model designed for morphology-aware reconstruction from indirect signals (Bedin & Coauthors, 2024). To quantify the contribution of each component of SE-Diff, we report ablations trained under identical schedules and seeds: (i) *SE-Diff w/o Sim* (removing the Euler consistency term $\mathcal{L}_{\text{Euler}}$); (ii) *SE-Diff w/o InterLead* (dropping $\mathcal{L}_{\text{inter-lead}}$); (iii) *SE-Diff w/o Exp* (disabling EHR retrieval and LLM distillation so conditioning uses only text+metadata).

### 4.1 ECG GENERATION RESULTS

We evaluate SE-Diff along four clinically aligned levels: *signal-level stability*, *feature-level physiology*, *diagnostic/semantic alignment*, and *beat-level morphology and interval fidelity*. At each level, we define the metrics and report aggregate results on both MIMIC-IV-ECG and PTB-XL.

**Signal-level Stability.** Given matched real and generated ECGs $(\mathbf{x}, \hat{\mathbf{x}})$ under the same condition $c$, we compute per-lead mean absolute error (MAE), normalized root mean squared error (NRMSE), and Pearson correlation $r$ to assess waveform fidelity and temporal consistency.

**Feature-level Physiology.** To evaluate preservation of basic physiology, we compare heart rate (HR) estimated from $\hat{\mathbf{x}}$ and $\mathbf{x}$ via the absolute error $\text{MAE}_{\text{HR}}$.

**Diagnostic Alignment.** We adopt a CLIP-style evaluation for ECG–text pairs: an ECG encoder $f_{\text{ecg}}(\cdot)$ and a text encoder $f_{\text{text}}(\cdot)$ produce $\ell_2$-normalized embeddings; cosine similarity quantifies alignment, $s(\mathbf{x}, \text{text}) = \langle f_{\text{ecg}}(\mathbf{x}), f_{\text{text}}(\text{text}) \rangle$. To control encoder bias, we report the relative CLIP score and the relative FID score:

$$\text{rCLIP} = \frac{s(\hat{\mathbf{x}}, \text{text})}{s(\mathbf{x}, \text{text})} \qquad \text{rFID} = \frac{\text{FID}(\mathcal{X}_r^{(1)}, \mathcal{X}_r^{(2)})}{\text{FID}(\hat{\mathcal{X}}, \mathcal{X}_r) + \text{FID}(\mathcal{X}_r^{(1)}, \mathcal{X}_r^{(2)})} \tag{26}$$

Distributional coverage/quality is measured with the Fréchet distance (FID) in the ECG embedding space, $\text{FID} = \|\mu_r - \mu_g\|_2^2 + \text{Tr}(\Sigma_r + \Sigma_g - 2(\Sigma_r \Sigma_g)^{1/2})$, where $(\mu_r, \Sigma_r)$ and $(\mu_g, \Sigma_g)$ denote the mean and covariance of real and generated ECG embeddings, $\hat{\mathcal{X}}$ denotes generated samples, and $\mathcal{X}_r^{(1)}, \mathcal{X}_r^{(2)}$ are disjoint splits of the real set. And we report its normalized relative score rFID.

Table 1: ECG generation performance on MIMIC-IV-ECG and PTB-XL datasets
.

| Model | MAE ↓ | NRMSE ↓ | $\text{MAE}_{HR}$ ↓ | rCLIP Score ↑ | rFID Score ↑ |
|---|---|---|---|---|---|
| **MIMIC-IV-ECG (internal)** | | | | | |
| SSSD | $0.4337 \pm 0.0300$ | $0.2027 \pm 0.0441$ | $27.37 \pm 14.84$ | $0.7213 \pm 0.0402$ | $0.9096 \pm 0.0398$ |
| WGAN | $0.1896 \pm 0.0605$ | $0.1301 \pm 0.0316$ | $31.54 \pm 2.15$ | $0.5688 \pm 0.0347$ | $0.5497 \pm 0.0192$ |
| BeatDiff | $0.7464 \pm 0.0070$ | $0.4756 \pm 0.0117$ | $27.74 \pm 1.49$ | $0.5167 \pm 0.0180$ | $0.8612 \pm 0.0039$ |
| DiffuSETS | $0.1092 \pm 0.0022$ | $0.0851 \pm 0.0012$ | $13.29 \pm 1.13$ | $0.9309 \pm 0.0036$ | $0.9209 \pm 0.0038$ |
| **SE-Diff** | $\mathbf{0.0923 \pm 0.0021}$ | $\mathbf{0.0714 \pm 0.0010}$ | $\mathbf{8.43 \pm 0.42}$ | $\mathbf{0.9470 \pm 0.0029}$ | $\mathbf{0.9509 \pm 0.0035}$ |
| w/o Exp | $0.0926 \pm 0.0022$ | $0.0730 \pm 0.0008$ | $15.06 \pm 0.34$ | $0.9099 \pm 0.0026$ | $0.9032 \pm 0.0062$ |
| w/o InterLead | $0.0934 \pm 0.0023$ | $0.0733 \pm 0.0006$ | $19.21 \pm 1.27$ | $0.9216 \pm 0.0029$ | $0.9128 \pm 0.0052$ |
| w/o Sim | $0.0965 \pm 0.0024$ | $0.0768 \pm 0.0014$ | $14.28 \pm 1.35$ | $0.9303 \pm 0.0041$ | $0.9138 \pm 0.0056$ |
| **PTB-XL (external)** | | | | | |
| SSSD | $0.6103 \pm 0.0204$ | $0.3818 \pm 0.0670$ | $15.22 \pm 11.51$ | $0.8618 \pm 0.0599$ | $0.7168 \pm 0.0355$ |
| WGAN | $0.2458 \pm 0.0653$ | $0.1197 \pm 0.0313$ | $13.82 \pm 18.69$ | $0.5880 \pm 0.0000$ | $0.5377 \pm 0.0232$ |
| BeatDiff | $0.9888 \pm 0.0059$ | $0.4731 \pm 0.0104$ | $13.86 \pm 0.78$ | $0.8799 \pm 0.0022$ | $0.8503 \pm 0.0035$ |
| DiffuSETS | $0.1281 \pm 0.0030$ | $0.0797 \pm 0.0011$ | $17.88 \pm 0.72$ | $0.8690 \pm 0.0011$ | $0.8456 \pm 0.0035$ |
| **SE-Diff** | $\mathbf{0.1076 \pm 0.0033}$ | $\mathbf{0.0630 \pm 0.0006}$ | $\mathbf{8.24 \pm 0.43}$ | $\mathbf{0.8901 \pm 0.0060}$ | $\mathbf{0.8583 \pm 0.0056}$ |
| w/o Sim | $0.1138 \pm 0.0032$ | $0.0680 \pm 0.0007$ | $14.72 \pm 0.90$ | $0.8896 \pm 0.0010$ | $0.8004 \pm 0.0061$ |
| w/o InterLead | $0.1084 \pm 0.0034$ | $0.0640 \pm 0.0007$ | $12.02 \pm 0.78$ | $0.7484 \pm 0.0076$ | $0.8568 \pm 0.0051$ |

Tables 1 report MAE, NRMSE, $\text{MAE}_{HR}$, rCLIP, rFID on MIMIC-IV-ECG and PTB-XL dataset. Across both datasets, SE-Diff consistently outperforms SSSD, WGAN, BeatDiff, and DiffuSETS on all metrics, indicating improved waveform reconstruction, more accurate heart rate estimation, and stronger ECG–text alignment. The ablations (SE-Diff w/o Sim, w/o InterLead, w/o Exp) each degrade one or more metrics, highlighting the importance of simulator guidance, inter-lead constraints, and experience-based conditioning.

**Beat-level Morphology and Interval Fidelity.** To assess beat-level fidelity beyond global signal errors and heart rate, we introduce a set of quantitative morphology and interval features. For each ECG, we automatically extract PR interval, QRS duration (QRSd), QT interval, heart-rate–corrected QT (QTcF), ST-segment deviation at J+60 ms (ST@J+60), and P- and T-wave durations (P dur, T dur), and compute record-level median absolute errors between generated and real ECGs.

Table 2: Beat-level morphology & interval fidelity (MAE) on MIMIC-IV-ECG and PTB-XL datasets. All values are record-level medians; lower is better (↓).

| Model | PR ↓ | QRSd ↓ | QT ↓ | QTcF ↓ | ST@J+60 ↓ | P dur ↓ | T dur ↓ |
|---|---|---|---|---|---|---|---|
| **MIMIC-IV-ECG (internal)** | | | | | | | |
| SSSD | $7.70 \pm 8$ | $32.00 \pm 1$ | $25.11 \pm 11$ | $23.25 \pm 11$ | $0.16 \pm 0$ | $5.30 \pm 3$ | $23.71 \pm 9$ |
| WGAN | $18.01 \pm 9$ | $29.64 \pm 9$ | $21.02 \pm 5$ | $23.83 \pm 2$ | $0.13 \pm 0$ | $6.01 \pm 0$ | $24.01 \pm 4$ |
| BeatDiff | $10.34 \pm 5$ | $18.23 \pm 6$ | $14.23 \pm 6$ | $9.43 \pm 5$ | $0.17 \pm 0$ | $7.00 \pm 1$ | $30.02 \pm 3$ |
| DiffuSETS | $14.81 \pm 6$ | $11.71 \pm 7$ | $8.20 \pm 3$ | $9.71 \pm 4$ | $0.04 \pm 0$ | $5.60 \pm 1$ | $8.71 \pm 7$ |
| **SE-Diff** | $\mathbf{7.30 \pm 3}$ | $\mathbf{10.71 \pm 4}$ | $\mathbf{4.50 \pm 2}$ | $\mathbf{7.88 \pm 2}$ | $\mathbf{0.03 \pm 0}$ | $\mathbf{2.50 \pm 0}$ | $\mathbf{6.80 \pm 3}$ |
| w/o InterLead | $11.11 \pm 3$ | $15.01 \pm 5$ | $11.91 \pm 5$ | $15.84 \pm 7$ | $0.03 \pm 0$ | $4.00 \pm 1$ | $8.10 \pm 4$ |
| w/o Exp | $9.81 \pm 9$ | $14.11 \pm 13$ | $8.20 \pm 3$ | $12.61 \pm 3$ | $0.04 \pm 0$ | $4.00 \pm 1$ | $6.90 \pm 3$ |
| w/o Sim | $12.11 \pm 7$ | $13.21 \pm 6$ | $8.40 \pm 4$ | $8.89 \pm 7$ | $0.04 \pm 0$ | $5.10 \pm 1$ | $13.11 \pm 1$ |
| **PTB-XL (external)** | | | | | | | |
| SSSD | $13.71 \pm 8$ | $13.61 \pm 8$ | $30.22 \pm 9$ | $27.69 \pm 14$ | $0.33 \pm 0$ | $12.71 \pm 5$ | $19.11 \pm 10$ |
| WGAN | $14.01 \pm 7$ | $17.45 \pm 13$ | $12.01 \pm 10$ | $12.97 \pm 10$ | $0.18 \pm 0$ | $7.33 \pm 3$ | $25.68 \pm 13$ |
| BeatDiff | $13.41 \pm 4$ | $17.40 \pm 6$ | $9.81 \pm 3$ | $8.94 \pm 2$ | $0.73 \pm 1$ | $4.10 \pm 1$ | $18.81 \pm 6$ |
| DiffuSETS | $7.70 \pm 2$ | $12.61 \pm 7$ | $11.51 \pm 4$ | $13.75 \pm 5$ | $0.10 \pm 0$ | $9.11 \pm 6$ | $13.20 \pm 5$ |
| **SE-Diff** | $\mathbf{3.90 \pm 1}$ | $\mathbf{10.01 \pm 2}$ | $\mathbf{5.20 \pm 1}$ | $\mathbf{8.68 \pm 2}$ | $\mathbf{0.07 \pm 0}$ | $\mathbf{3.00 \pm 1}$ | $\mathbf{9.31 \pm 8}$ |
| w/o InterLead | $7.00 \pm 1$ | $10.31 \pm 5$ | $8.40 \pm 4$ | $10.91 \pm 3$ | $0.07 \pm 0$ | $4.40 \pm 1$ | $13.61 \pm 8$ |
| w/o Sim | $9.21 \pm 4$ | $12.91 \pm 5$ | $12.51 \pm 4$ | $14.20 \pm 2$ | $0.07 \pm 0$ | $4.50 \pm 1$ | $12.31 \pm 3$ |

Tables 2 summarize the beat-level morphology and interval metrics on MIMIC-IV-ECG and PTB-XL. SE-Diff achieves the lowest median errors across all intervals (PR, QRSd, QT/QTcF, ST@J+60, P dur, T dur), demonstrating that it not only matches global signal statistics but also preserves beat-level timing and morphology more faithfully than competing models.

## 4.2 DOWNSTREAM ECG CLASSIFICATION

We evaluate whether SE-Diff mitigates severe class imbalance in downstream ECG classification by augmenting minority classes with model-generated ECGs. Training distributions are intentionally skewed; evaluation uses a fixed, class-balanced test set. We compare four regimes: (i) *Unbalanced* (real-only, skewed), (ii) *Balanced* (real-only, fully balanced; reference upper bound), (iii) several synthetic augmentation strategies that oversample the minority class using SSSD, WGAN, BeatDiff, DiffuSETS, and (iv) SE-Diff. We report F1, accuracy (Acc.), and AUROC (AUC).

**Imbalanced Gender Classification.** We train a binary classifier to predict sex (Female vs. Male). As shown in Table 3, all generative augmentation methods improve over the *Unbalanced* baseline, but SE-Diff yields the largest gains, raising F1 from $42\%$ to $58\%$ and AUC from $46\%$ to $58\%$. This substantially narrows the gap to the *Balanced* upper bound (F1/AUC $62\%$), and SE-Diff consistently outperforms SSSD, WGAN, BeatDiff, and DiffuSETS under the same skew.

**Rare-disease Classification.** We train a classifier to distinguish Sinus rhythm from supraventricular tachycardia (SVT), treating SVT as the minority class. Table 3 shows larger relative gains on SVT, indicating that synthetic augmentation is particularly effective when physiological heterogeneity is high and labeled minority examples are scarce. SE-Diff recovers a substantial fraction of this gap, especially on the minority SVT class.

Table 3: Downstream ECG classification under severe class imbalance.

| Model | Male = 10% Female | | | SVT = 10% Sinus | | |
|---|---|---|---|---|---|---|
| | F1 (%, ↑) | Acc. (%, ↑) | AUC (%, ↑) | F1 (%, ↑) | Acc. (%, ↑) | AUC (%, ↑) |
| SSSD | $42 \pm 0$ | $54 \pm 1$ | $49 \pm 2$ | $56 \pm 1$ | $63 \pm 2$ | $81 \pm 2$ |
| WGAN | $42 \pm 0$ | $54 \pm 1$ | $49 \pm 2$ | $57 \pm 2$ | $63 \pm 2$ | $82 \pm 2$ |
| BeatDiff | $44 \pm 2$ | $55 \pm 2$ | $51 \pm 2$ | $60 \pm 2$ | $67 \pm 2$ | $84 \pm 3$ |
| DiffuSETS | $44 \pm 3$ | $54 \pm 2$ | $54 \pm 1$ | $70 \pm 1$ | $68 \pm 1$ | $84 \pm 2$ |
| **SE-Diff** | $\mathbf{58 \pm 1}$ | $\mathbf{58 \pm 1}$ | $\mathbf{58 \pm 2}$ | $\mathbf{72 \pm 2}$ | $\mathbf{71 \pm 0}$ | $\mathbf{85 \pm 2}$ |
| Unbalanced | $42 \pm 0$ | $54 \pm 1$ | $46 \pm 2$ | $56 \pm 1$ | $62 \pm 0$ | $80 \pm 1$ |
| Balanced | $62 \pm 0$ | $62 \pm 0$ | $62 \pm 1$ | $79 \pm 1$ | $80 \pm 2$ | $93 \pm 1$ |

## 4.3 MECHANISTIC ANALYSIS OF SE-DIFF

**Noise Scheduling Analysis.** Figure 2 summarizes the forward process under our linear noise schedule. Panel (a) shows the per-step noise increment increasing steadily, while (b) displays the corresponding signal retention factor decreasing slightly each step. The cumulative signal fraction in (c) drops smoothly from near one to near zero, and the accumulated noise in (d) rises monotonically and saturates in late timesteps. This profile yields a gradual, well-conditioned reverse trajectory: early steps recover global rhythm and cross-lead coherence, and later steps refine P/QRS/T morphology and suppress residual artifacts. We therefore adopt this schedule for SE-Diff as it offers an interpretable progression and stable behavior across sampling budgets.

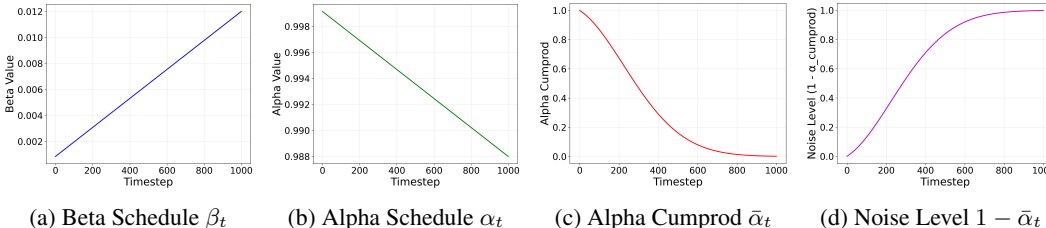

(a) Beta Schedule $\beta_t$     (b) Alpha Schedule $\alpha_t$     (c) Alpha Cumprod $\bar{\alpha}_t$     (d) Noise Level $1 - \bar{\alpha}_t$

Figure 2: Noise scheduling analysis showing the progression of noise and signal factors throughout the diffusion process.

**Case Study for ECG Simulator.** To visualize the simulator's morphology, Fig. 3 presents four random single-cycle templates (one lead per label): *(A) Sinus rhythm, Lead I.* Upright P wave, narrow QRS complex, and concordant T wave provide a clean normal reference for comparison. *(B) Ventricular pacing, Lead V1.* A wide, predominantly negative QRS complex (QS/deep S), reflecting pacing/LBBB-like activation, clearly departs from normal conduction. *(C) Sinus rhythm with*

*first-degree AV block, Lead II.* A P wave followed by an elongated isoelectric segment before the QRS complex qualitatively indicates PR-interval prolongation. *(D) Consider acute ST-elevation MI, Lead V3.* Convex ST-segment elevation after the J point is characteristic of anteroseptal involvement. The simulator serves as a morphology prior and qualitative oracle within SE-Diff, enabling clear visual audits and morphology-aware ablations without confounding rhythm variability, and providing guidance to the diffusion model.

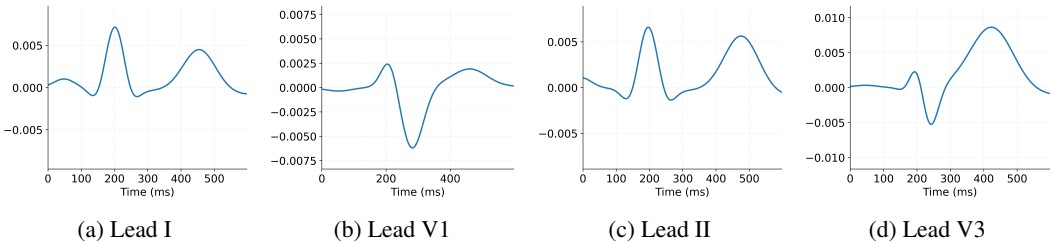

| (a) Lead I | (b) Lead V1 | (c) Lead II | (d) Lead V3 |

Figure 3: Representative single-cycle ECG waveforms generated from our simulator. Panel A: sinus rhythm (Lead I). Panel B: ventricular pacing (Lead V1). Panel C: sinus rhythm with first-degree AV block (Lead II). Panel D: consider acute ST-elevation MI (Lead V3).

**Case Study for ECG Generation.** Figure 4 compares a 10 s, 12-lead ECG generated by SE-Diff with its paired reference for a case conditioned on *"Sinus rhythm"* (male, 65 y, HR 94 bpm). The generation preserves canonical sinus morphology—each $P$ wave preceding a narrow QRS complex with an appropriate PR interval—and shows coherent R-wave progression across the precordial leads, with R–R intervals consistent with the target rate. Clinically, the SE-Diff tracing appears cleaner than the reference: baseline wander and high-frequency artifacts are attenuated, yielding crisper ST segments and T-wave contours without distorting morphology. This qualitative finding aligns with the model's design: simulator-informed constraints and experience-augmented conditioning steer the diffusion process toward physiologically plausible, low-noise signals.

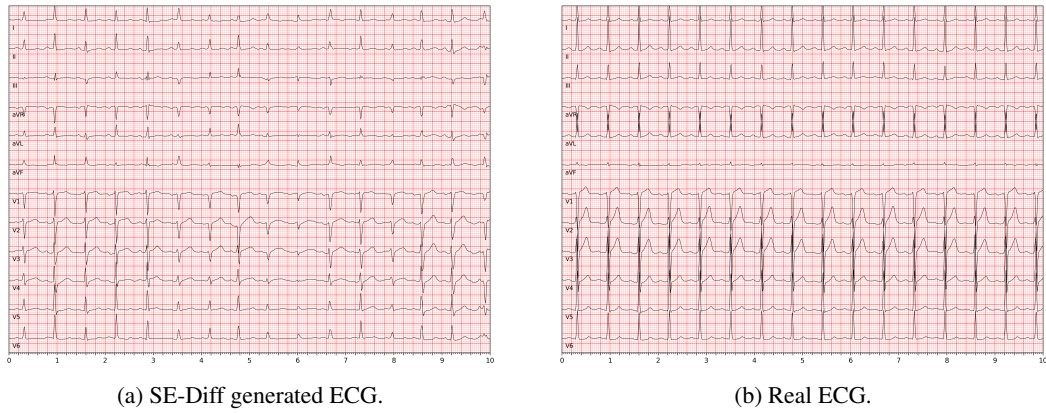

(a) SE-Diff generated ECG.    (b) Real ECG.

Figure 4: Case Study for ECG Generation.

## 5 CONCLUSION

We introduced SE-Diff, a conditional latent-diffusion framework for 10-second, 12-lead ECG synthesis that couples a VAE latent space with a Beat Decoder and simulator-informed regularizers, and strengthens conditioning via experience retrieval from EHRs. Across benchmarks, SE-Diff improves signal fidelity, preserves inter-lead physiology, and achieves tighter diagnostic/semantic alignment, while also enhancing downstream classification when used for data augmentation. Ablations confirm that both the ODE-based guidance (Euler and inter-lead constraints) and retrieval-augmented conditioning contribute materially to performance. Future work will extend SE-Diff to more clinically meaningful applications (e.g., arrhythmia risk stratification, therapy response modeling, and long-term ambulatory ECG) and evaluate robustness across institutions and rare presentations. We believe SE-Diff offers a principled step toward physiologically grounded, clinically aligned generative modeling of ECGs.

## 6 ACKNOWLEDGEMENT

This research was partially supported by the U.S. National Science Foundation under Award Numbers 2442172, 2312502, 2319449, 2211557, 2303037, 2312501, 2531008, and 2106859, and by the U.S. National Institutes of Health under Award Numbers K25DK135913, RF1NS139325, R01DK143456, U18DP006922, OT2OD038003, R01HL175135, U54OD036472, U54HG012517, and U24DK097771. This research was also partially supported by internal funds and GPU servers provided by the Computer Science Department of Emory University, the SRC JUMP 2.0 Center, Amazon Research Awards, Snapchat Gifts, Optum AI, NEC, and the Easton Center.

## 7 ETHICS STATEMENT

We adhere to the ICLR Code of Ethics. This study uses only de-identified data from MIMIC-IV-ECG (Gow et al., 2023; Johnson et al., 2023) and associated de-identified records from MIMIC-IV-CLINICAL (Johnson et al., 2023) under the applicable Data Use Agreements and credentialed-access requirements. No direct interaction with human subjects occurred; no personally identifiable information (PII/PHI) was accessed or released, and we made no attempt at re-identification. Heart-rate estimation (WFDB XQRS), resampling, and VAE-based encoding were performed on the de-identified waveforms; EHR linkage relied only on the dataset's de-identified subject and admission keys within documented admission windows. This work is for research purposes only and is not a medical device. No conflicts of interest or sensitive sponsorships are present.

## 8 REPRODUCIBILITY STATEMENT

All information necessary to reproduce our results is documented in Appendix C, including dataset curation and preprocessing, model architectures, training and inference hyperparameters and schedules, and so on.

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

## A    USE OF LARGE LANGUAGE MODELS

We used a large language model (LLM) solely for writing assistance—specifically, to correct grammar, improve wording, and enhance clarity. The LLM did not contribute to research design, data analysis, modeling, experiments, or interpretation of results. All technical content and conclusions were authored and verified by the authors, who take full responsibility for the manuscript.

### A.1    RELATED WORK

#### A.1.1    GENERATIVE MODELS FOR ECG

Simulation has repeatedly improved data efficiency in sequential decision making—both in imitation learning and in reinforcement learning—by narrowing the gap between training and deployment (Zadok et al., 2019; Mnih et al., 2013; Ni et al., 2025; Zeng et al., 2025). In parallel, generative models have been used to expand training corpora: in vision, SimGAN refines synthetic images with unlabeled real data (Shrivastava et al., 2017); in cardiology, augmenting classifiers with GAN-generated heartbeats improves performance (Golany et al., 2020a). Beyond beat-level augmentation, adversarial models have produced realistic multi-lead "DeepFake" ECGs for privacy and data scarcity mitigation (Thambawita et al., 2021), and mechanism-aware variants embed ordinary differential equations to better capture depolarization–repolarization dynamics (Golany et al., 2021). However, GANs can be unstable and prone to mode collapse in multi-lead, multi-label

regimes. Denoising diffusion and score-based models offer a likelihood-grounded alternative with strong mode coverage and stable training (Ho et al., 2020; Song et al., 2021). Recent ECG adaptations include conditional diffusion with structured state-space backbones (SSSD-ECG) (Lopez Alcaraz & Strodthoff, 2023), generalized diffusion for generation/imputation/forecasting (Neifar et al., 2023), state-space/transformer hybrids (Zama & Schwenker, 2023), and text/metadata-conditioned synthesis (DiffuSETS) (Lai et al., 2025a). The field is also trending toward personalization and physiological consistency: conditional models incorporate patient metadata or anatomy to produce more plausible 12-lead signals (Sang et al., 2025), and diffusion frameworks create patient "digital twins" (Lai et al., 2025b). Hybrid uses couple generative modeling with signal-quality assessment and anomaly detection (Han et al., 2025), while semi-supervised GANs aim to better capture temporal dynamics (Li et al., 2025). Very recent work explores flow-matching as a faster alternative to iterative diffusion for ECG synthesis, reducing sampling cost while targeting comparable fidelity (Bondar et al., 2025).

### A.1.2 PHYSIOLOGICAL ECG SIMULATORS

Compact physiological simulators capture stereotyped P–QRS–T morphology with low-dimensional differential equations. The canonical ECGSYN model uses a three-dimensional limit-cycle oscillator whose phase-locked Gaussian components generate P, QRS, and T deflections, while stochastic control of instantaneous heart rate reproduces realistic RR patterns and HRV statistics (e.g., mean/SD of RR, low- and high-frequency spectral peaks) (McSharry et al., 2003; Task Force of the European Society of Cardiology and the North American Society of Pacing and Electrophysiology, 1996). Open implementations (e.g., PhysioNet ECGSYN) enable reproducible waveform synthesis and stress-testing (Goldberger et al., 2000). However, globally fixed morphology templates and linear lead projections limit expressivity under rhythm changes, conduction abnormalities, and nonstationary repolarization. Hybrid approaches mitigate these issues by coupling mechanistic priors with learnable components—via neural ODEs or universal differential equations—to preserve physical structure while fitting data (Han et al., 2024); conditioning on anatomy further improves inter-lead realism (Chen et al., 2018; Rackauckas et al., 2021; Sang et al., 2025).

### A.1.3 ECG CLASSIFICATION

Classical ECG pipelines segment signals into beats with robust QRS detectors (e.g., Pan–Tompkins; Afonso et al.) and derive interval/morphology descriptors before applying shallow classifiers such as linear discriminants or SVMs (Afonso et al., 1999; De Chazal et al., 2004; Nasrabadi, 2007). With deep learning, end-to-end models on raw waveforms supplanted hand-crafted features and reached cardiologist-level performance in single- and ambulatory-lead arrhythmia detection (Rajpurkar et al., 2017; Hannun et al., 2019); at the beat level, residual CNNs are particularly effective, and large multi-lead corpora such as PTB-XL have enabled high-capacity models and rigorous multi-label benchmarking (Kachuee et al., 2018; Wagner et al., 2020). Recent work refines architectures and training regimes—dual-channel networks that fuse ResNet-ICBAM with 2D-CNN features emphasize region-of-interest cues (Wang et al., 2025a), ECG-specific scaling laws suggest shallower but wider networks outperform vision-oriented designs (Lee et al., 2023), and transfer learning on transformed signals improves performance under class imbalance (Mavaddati, 2025). Synthetic data from GANs and diffusion models is now routinely used for augmentation, with semi-supervised variants further boosting diagnostic accuracy (Li et al., 2025). In our experiments, we adopt a strong ResNet heartbeat classifier and evaluate whether SE-Diff improves generalization under class imbalance and limited labels by augmenting training with physiologically plausible, label-consistent synthetic beats.

## B ECG SIMULATOR CALIBRATION WITH STABILIZERS AND MORPHOLOGY PRIORS

**Motivation.** Naive least-squares calibration of the ODE-based simulator (Sec. 2.2) often fits the sharp QRS complex yet drifts later in the window and may flip polarity. The main causes are small baseline trends and scale mismatches between simulated and observed signals, and an under-constrained morphology (especially T-wave width). We introduce lightweight, differentiable sta-

bilizers that improve convergence and yield physiologically plausible parameters without altering simulator dynamics.

**Trend-aware alignment and fidelity.** Let $z_\eta(t)$ denote the simulated voltage with parameters $\eta$. Rather than compare $z_\eta$ directly to the observation $y(t)$, align via a three-parameter affine–trend model with scale $s$, offset $c$, and linear trend $b$:

$$\hat{y}(t) = c + s\, z_\eta(t) + b\big(t - \bar{t}\big), \qquad \bar{t} = \tfrac{1}{T} \sum_{t=1}^{T} t. \tag{27}$$

At each iteration $(c, s, b)$ are obtained by least squares and are differentiable in $z_\eta$. The fidelity term is

$$\mathcal{L}_{\text{mse}}(\eta) = \tfrac{1}{T} \sum_{t=1}^{T} \big(\hat{y}(t) - y(t)\big)^2, \tag{28}$$

augmented by a small scale regularizer to prevent rare runaway gains:

$$\mathcal{L}_s(\eta) = \lambda_s\, s^2, \qquad \lambda_s \in [10^{-6}, 10^{-5}]. \tag{29}$$

**Morphology priors (widths and amplitudes).** Using the McSharry parameterization, each deflection $\beta \in \{P, Q, R, S, T\}$ has amplitude $a_\beta$, width $b_\beta > 0$, and phase $\theta_\beta$. Enforce positivity with $b_\beta = \text{softplus}(\tilde{b}_\beta) + \varepsilon$ $(\varepsilon = 10^{-3})$ and shrink widths toward physiological targets $b_\beta^\star$:

$$\mathcal{L}_{\text{width}}(\eta) = \lambda_b \sum_\beta w_\beta \big(b_\beta - b_\beta^\star\big)^2, \quad (b_P^\star, b_Q^\star, b_R^\star, b_S^\star, b_T^\star) = (0.20, 0.08, 0.10, 0.08, 0.32), \tag{30}$$

with $w_T = 2$ and $w_\beta = 1$ otherwise to prevent absorbing baseline drift via an overly broad T wave. A mild amplitude penalty

$$\mathcal{L}_{\text{amp}}(\eta) = \lambda_a \sum_\beta a_\beta^2 \tag{31}$$

discourages attributing variability solely to the global scale $s$ in Eq. 27.

**Phase ordering.** To preserve the physiological ordering of $\{P, Q, R, S, T\}$ on the unit circle, introduce a global phase shift $\Delta\theta$ and wrap phases as $\theta_\beta \leftarrow \text{wrap}(\tilde{\theta}_\beta + \Delta\theta)$. A hinge penalty with margin $m$ enforces monotonicity:

$$\mathcal{L}_{\text{ord}}(\eta) = \lambda_{\text{ord}} \sum_{i=1}^{4} \max\big\{0,\, \theta_i - \theta_{i+1} + m\big\}, \qquad m \approx 0.05 \text{ rad}. \tag{32}$$

This term typically decays after a few epochs and can be disabled once ordering stabilizes.

**Objective and optimization.** The calibration loss is

$$\mathcal{L}(\eta) = \mathcal{L}_{\text{mse}} + \mathcal{L}_s + \mathcal{L}_{\text{width}} + \mathcal{L}_{\text{amp}} + \mathcal{L}_{\text{ord}}. \tag{33}$$

Optimize with AdamW (cosine decay with warmup), followed by a brief L-BFGS refinement. The same Euler sub-stepping and burn-in used at inference are applied during training to maintain integrator consistency.

**Polarity canonicalization.** Because lead inversions are common, canonicalize polarity after fitting: re-simulate $z_\eta$, re-estimate $\hat{y}(t) = c + s\, z_\eta(t)$ (no slope), and, if $s < 0$, flip all amplitudes $\{a_\beta\}$ once. This step is outside the loss and standardizes reported parameters.

## C   IMPLEMENTATION DETAILS.

All models are trained in PyTorch with AMP on a single NVIDIA H200 using AdamW (lr $1 \times 10^{-4}$ with cosine decay to $1 \times 10^{-5}$), gradient clipping/accumulation (global batch 4096), for 200 epochs with early stopping. Diffusion uses $T{=}1000$ steps, linear $\beta_t \in [8.5 \times 10^{-4}, 1.2 \times 10^{-2}]$ (`DDPMScheduler`), and classifier-free guidance. The VAE has 4 latent channels; the encoder/decoder are multi-resolution with residual blocks, attention, and skip connections; training uses $\mathcal{L}_{\text{MSE}} + \mathcal{L}_{\text{KL}}$ ($\lambda_{\text{KL}}{=}1$). A lightweight beat decoder predicts the first beat ($L_c{=}300$ at 500 Hz;

R-peaks via `NeuroKit2`). The denoiser is a 7-stage 1D U-Net (kernel 7) with self/cross-attention (8 heads, width 16–64) consuming text embeddings (1536-d) plus metadata (age, sex, heart rate). Physiology-aware training adds the Euler simulator loss ($\lambda{=}3{\times}10^{-3}$) and the inter-lead constraint ($\gamma{=}5{\times}10^{-2}$); class-wise simulator parameters $\eta$ are prefit from 200 beats/label.

We use MIMIC-IV-ECG with simplified rhythm labels. Free-text diagnostic reports are cleaned and normalized, then mapped to a compact multi-label taxonomy (e.g., sinus rhythm/brady/tachy, atrial fibrillation/flutter, PAC/PVC, bundle-branch block, LVH/RVH, prolonged QT, ST/T abnormalities, ischemia/infarct). Texts are embedded with a pretrained text encoder (`text-embedding-ada-002`). For simulator-informed diffusion, we pre-compute class-wise simulator parameters for the top-20 ECG categories and use them during training. For ECG generation, we sample 100 waveforms per setting and compute metrics. For downstream ECG classification, we form balanced subsets with 200 samples per class and train a lightweight MLP that flattens latents (512-d) and applies two fully connected layers (128→64) with BatchNorm, ReLU, and Dropout (0.5), followed by a linear output; optimization uses cross-entropy with AdamW and early stopping. In all experiments, we set $\lambda_1 = \lambda_2 = \lambda_3 = 1$ and use $k = 3$ retrieved admissions for experience-based conditioning. And we employ the 'gpt-4.1-mini' model for generating the experience report $r$. The Beat Decoder is trained jointly with the VAE, and when R-peak detection fails or no class-wise simulator parameters are available for a given sample, we simply skip the beat/spectral or corresponding simulator losses for that sample to avoid instability while keeping the rest of the training unchanged.

From an ethics and privacy perspective, all experiments are conducted on de-identified public datasets (MIMIC-IV-ECG/EHR and PTB-XL), and we enforce strict *patient-level* splits: records used at validation/test time are never included in the retrieval pool, so the EHR retrieval context cannot contain labels or text from the same patient. Retrieval is restricted to the training cohort only. Moreover, the simulator-informed and experience-augmented model maintains its advantage on PTB-XL as an external cohort, suggesting that SE-Diff provides utility without simply memorizing individual training cases or overfitting to specific MIMIC records.

## D  PROMPT EXAMPLE

The prompt example in our SE-Diff can be shown in Figure 5.

> === SYSTEM PROMPT ===
>
> You are a cardiology-focused language model. Your job is to convert structured inputs into a concise, clinically accurate narrative suitable for a cardiology note. You must not invent findings and you must not provide treatment recommendations. You should prioritize the current patient's measurements and observations over any retrieved examples from similar patients.
>
> The report you generate will serve as the conditioning text for a downstream text-to-ECG synthesis model. Therefore, the narrative must be internally consistent, and physiologically grounded so that it can guide waveform generation.
>
> You will receive (1) current patient context, (2) the original machine generated diagnosis, and (3) retrieval diagnosis from clinically similar patients. Based on these inputs, you will produce exactly one paragraph of 60-120 words. Your language should be precise and use standard ECG terminology and units so the downstream generator can align text with waveform features.
>
> === DATA PROMPT ===
>
> Original machine-generated diagnosis (most important first): Sinus tachycardia | Normal ECG except for rate
> Heart rate (bpm): 100
> Age (years): 52
> Gender: F
> Retrieved similar patients' diagnoses: Sinus rhythm | Normal ECG | Possible ectopic atrial tachycardia

Figure 5: Prompt examples.

# E  CONDITIONAL LATENT DIFFUSION—IMPLEMENTATION DETAILS

*Context construction.* The context $c$ concatenates (i) token embeddings from the clinical report $E_{\text{text}} \in \mathbb{R}^{m \times d_c}$ (from a frozen clinical text encoder) and (ii) a single metadata token $e_{\text{meta}} \in \mathbb{R}^{1 \times d_c}$ formed by projecting age/sex/*etc.* to $d_c$. We set $C = [E_{\text{text}}; e_{\text{meta}}] \in \mathbb{R}^{(m+1) \times d_c}$ and feed $C$ to cross-attention at the bottleneck and (optionally) the two highest-resolution decoder blocks.

*U-Net blocks.* Each block consists of Conv($k{=}3$) $\to$ GroupNorm $\to$ SiLU $\to$ Conv($k{=}3$) with a residual connection. Self-attention (multi-head) follows normalization at lower resolutions. Down-sampling uses stride-2 convolutions; upsampling uses nearest-neighbor followed by Conv($k{=}3$). We use $K$ resolution levels (e.g., $K{=}4$) and $N$ blocks per level (e.g., $N{=}2$); channel width doubles on downsampling and halves on upsampling.

*Attention.* Cross-/self-attention are multi-head with head dimension $d_a$ (e.g., 8 heads). Queries at the bottleneck attend to keys/values derived from $C$; text tokens use sinusoidal positional encodings. Metadata is represented as a single learned token (Huang et al., 2019; Liu et al., 2025).

*Time embedding and FiLM.* The timestep embedding uses exponentially spaced sinusoids and a two-layer MLP (SiLU) to produce $(\gamma_t, \beta_t)$ per block. A small MLP on $\text{Pool}(C)$ (token average) yields $(\gamma_c, \beta_c)$. FiLM is applied after normalization to every residual block.

*Training setup.* We use a cosine noise schedule and optional learned variance (Nichol & Dhariwal, 2021), AdamW with EMA, and classifier-free guidance with unconditional dropout $p_{\text{uncond}} \in [0.1, 0.2]$ and tuned guidance scale. Min-SNR-$\gamma$ weighting (Hang et al., 2023) is optional for stabilizing early and late timesteps. Simulator losses are computed on $h = D_\psi^{\text{beat}}(z_0)$ and removed at inference.

# F  ABLATION STUDY OF BEAT AND SPECTRAL LOSSES ON THE BEAT DECODER

To further clarify the relationship between the time-domain beat reconstruction loss $L_{\text{beat}}$ in Eq. (13) and the spectral consistency loss $L_{\text{spec}}$ in Eq. (15), we conduct an ablation study where we remove each term in turn while keeping the rest of the architecture and training protocol unchanged. $L_{\text{beat}}$ encourages the Beat Decoder output $h$ to match a single R-peak–centered cycle $C(x)$ in the time domain, whereas $L_{\text{spec}}$ matches the log-magnitude spectrum of $h$ to the *distribution* of spectra computed over all detected beats $\{C_j(x)\}_{j=1}^{J}$ in the same 10 s window. The two objectives therefore act on complementary aspects of the canonical beat: $L_{\text{beat}}$ anchors $h$ to an actual observed cycle, while $L_{\text{spec}}$ regularizes its frequency content toward the ensemble of beats rather than the full 10 s signal.

Table 4 reports ECG generation performance on MIMIC-IV-ECG when using both losses (full SE-Diff), removing $L_{\text{beat}}$ ("w/o Beat Loss"), and removing $L_{\text{spec}}$ ("w/o Spec Loss"). Using both losses yields the best performance across all metrics. In particular, dropping $L_{\text{beat}}$ leads to a drastic degradation in heart-rate accuracy ($\text{MAE}_{\text{HR}}$ increases from 8.43 to 23.48) and noticeably worse rCLIP and rFID scores, indicating that the canonical beat is no longer well aligned with the temporal structure of the underlying rhythm. Removing $L_{\text{spec}}$ instead mainly harms signal-level fidelity (higher NRMSE and weaker rCLIP/rFID), showing that spectral regularization is important for capturing realistic morphology and fine-scale waveform details.

These results empirically support that $L_{\text{beat}}$ and $L_{\text{spec}}$ are *complementary* rather than conflicting: the best-performing model is obtained when the canonical beat $h$ is simultaneously constrained to resemble an actual beat in the time domain and to share spectral statistics with the set of beats present in the 10 s record.

Table 4: ECG generation performance on MIMIC-IV-ECG dataset.

| Model | MAE $\downarrow$ | NRMSE $\downarrow$ | $\text{MAE}_{\text{HR}}$ $\downarrow$ | rCLIP Score $\uparrow$ | rFID Score $\uparrow$ |
|---|---|---|---|---|---|
| **SE-Diff** | **0.0923 $\pm$ 0.0021** | **0.0714 $\pm$ 0.0010** | **8.43 $\pm$ 0.42** | **0.9470 $\pm$ 0.0029** | **0.9509 $\pm$ 0.0035** |
| w/o Beat Loss | 0.0951 $\pm$ 0.0020 | 0.0743 $\pm$ 0.0016 | 23.48 $\pm$ 1.07 | 0.9393 $\pm$ 0.0026 | 0.9140 $\pm$ 0.0055 |
| w/o Spec Loss | 0.0942 $\pm$ 0.0023 | 0.0797 $\pm$ 0.0014 | 16.47 $\pm$ 1.23 | 0.9257 $\pm$ 0.0030 | 0.9073 $\pm$ 0.0055 |

## G INFERENCE EFFICIENCY OF SE-DIFF FOR 10-SECOND, 12-LEAD ECG GENERATION

To show the inference efficiency, we report the wall-clock latency of SE-Diff for generating 10 s, 12-lead ECGs in Table 5. For each batch size, we measure the time required to run 1000 diffusion sampling steps and average the runtime over three batches.

The results show that the total time per batch remains nearly constant for batch sizes between 32 and 256, so the per-sample latency decreases from 0.425 s at batch size 32 to 0.050 s at batch size 256. Even for very large batches (up to 4096), the wall-clock time grows sublinearly with the batch size, and the per-sample latency can be reduced to 0.0205 s. This indicates that SE-Diff can efficiently exploit batching during inference, making large-scale ECG synthesis practical under our current implementation.

Table 5: Inference latency of the model for generating one 10-second, 12-lead ECG under different batch sizes. Times are averaged over 3 batches per configuration, with 1000 diffusion sampling steps.

| Batch size | Time / batch (s) | Time / sample (s) |
|---|---|---|
| 32 | 13.55 | 0.425 |
| 128 | 13.65 | 0.105 |
| 256 | 13.40 | 0.050 |
| 512 | 15.05 | 0.0295 |
| 1024 | 24.90 | 0.0245 |
| 2048 | 45.00 | 0.0220 |
| 4096 | 84.75 | 0.0205 |

## H EFFECTS OF NOISE SCHEDULE AND SAMPLING STEPS

We first study how the number of diffusion sampling steps affects ECG generation quality when keeping the sampler (DDPM) and noise schedule (linear) fixed. As shown in Table 6, performance generally improves as we increase the number of steps: reducing the steps from 1000 to 800 and 500 leads to progressively higher NRMSE, larger heart-rate error ($\text{MAE}_{\text{HR}}$), and lower rCLIP scores. The default setting of 1000 steps achieves the best trade-off among these metrics, and is therefore used in our main experiments; smaller step counts can still be used in latency-constrained scenarios at the cost of degraded fidelity and semantic alignment.

Table 6: ECG generation performance under different sampling steps.

| Sampling Steps | NRMSE ↓ | $\text{MAE}_{\text{HR}}$ ↓ | rCLIP Score ↑ |
|---|---|---|---|
| 1000 | **0.0714 ± 0.0010** | **8.43 ± 0.42** | **0.9470 ± 0.0029** |
| 800 | 0.0877 ± 0.0011 | 13.59 ± 1.05 | 0.8671 ± 0.0040 |
| 500 | 0.1097 ± 0.0010 | 15.60 ± 1.19 | 0.7888 ± 0.0084 |

Next, we compare different sampler types while fixing the number of sampling steps (1000) and using a linear noise schedule. Table 7 shows that our default DDPM sampler in SE-Diff achieves the best overall balance across $\text{MAE}_{\text{HR}}$, and rCLIP. While the SDE-based sampler attains slightly lower NRMSE, it substantially worsens heart-rate accuracy and semantic alignment (higher $\text{MAE}_{\text{HR}}$ and lower rCLIP). DDIM and the second-order solver also underperform DDPM in at least one key metric. These results support our choice of DDPM as the default sampler for ECG generation in SE-Diff.

Finally, we investigate the impact of the noise schedule under the same DDPM sampler and 1000 sampling steps. As reported in Table 8, both the linear and cosine schedules yield comparable performance, but the linear schedule achieves slightly better NRMSE, $\text{MAE}_{\text{HR}}$, and rCLIP. This confirms that our default linear schedule is a strong and stable choice for ECG generation in SE-Diff, while alternative schedules can be used if desired without dramatically changing performance.

Table 7: ECG generation performance under different samplers.

| Sampler | NRMSE ↓ | MAE$_{HR}$ ↓ | rCLIP Score ↑ |
|---|---|---|---|
| **SE-Diff** (DDPM) | $0.0714 \pm 0.0010$ | $\mathbf{8.43 \pm 0.42}$ | $\mathbf{0.9470 \pm 0.0029}$ |
| DDIM | $0.1219 \pm 0.0010$ | $9.05 \pm 0.44$ | $0.7783 \pm 0.0072$ |
| Second Order | $0.0877 \pm 0.0011$ | $13.59 \pm 1.05$ | $0.8671 \pm 0.0040$ |
| SDE | $\mathbf{0.0693 \pm 0.0106}$ | $19.21 \pm 1.27$ | $0.9213 \pm 0.0028$ |

Table 8: ECG generation performance under different noise schedules.

| Sampler | NRMSE ↓ | MAE$_{HR}$ ↓ | rCLIP Score ↑ |
|---|---|---|---|
| **SE-Diff** (Linear) | $\mathbf{0.0714 \pm 0.0010}$ | $\mathbf{8.43 \pm 0.42}$ | $\mathbf{0.9470 \pm 0.0029}$ |
| Cosine | $0.0728 \pm 0.0032$ | $8.90 \pm 0.78$ | $0.9446 \pm 0.0131$ |

## I    ROBUSTNESS OF rCLIP AND rFID TO ENCODER CHOICES

To assess the potential dependence of rCLIP and rFID on the particular ECG/text encoder pair, we compare SE-Diff and DiffuSETS under three encoder configurations. **Config A** uses our baseline ECG and text encoders. **Config B** replaces the ECG encoder with an alternative ECG representation while keeping the text encoder fixed. **Config C** instead replaces the text encoder while keeping the ECG encoder fixed. For each configuration, we jointly report rCLIP, rFID, signal-level reconstruction error (MAE), and heart-rate error (MAE$_{HR}$).

As shown in Table 9, SE-Diff consistently outperforms DiffuSETS across *all* encoder configurations and metrics: MAE and MAE$_{HR}$ are always lower for SE-Diff, and both rCLIP and rFID are always higher. Moreover, the relative ranking between SE-Diff and DiffuSETS is stable when switching ECG or text encoders, indicating that our conclusions are not tied to a specific encoder choice. We also observe that improvements in rCLIP/rFID co-occur with gains in heart-rate and signal-level fidelity, suggesting that these representation-based metrics are aligned with clinically meaningful waveform quality rather than being purely encoder-specific artifacts.

Table 9: ECG generation performance comparison between SE-Diff and DiffuSETS under different calibration configurations.

| Config | Model | MAE ↓ | MAE$_{HR}$ ↓ | rCLIP Score ↑ | rFID Score ↑ |
|---|---|---|---|---|---|
| A | DiffuSETS | $0.1092 \pm 0.0022$ | $13.29 \pm 1.13$ | $0.9309 \pm 0.0036$ | $0.9209 \pm 0.0038$ |
| A | **SE-Diff** | $\mathbf{0.0923 \pm 0.0021}$ | $\mathbf{8.43 \pm 0.42}$ | $\mathbf{0.9470 \pm 0.0029}$ | $\mathbf{0.9509 \pm 0.0035}$ |
| B | DiffuSETS | $0.1092 \pm 0.0022$ | $13.29 \pm 1.13$ | $0.9347 \pm 0.0032$ | $0.9197 \pm 0.0037$ |
| B | **SE-Diff** | $\mathbf{0.0923 \pm 0.0021}$ | $\mathbf{8.43 \pm 0.42}$ | $\mathbf{0.9472 \pm 0.0032}$ | $\mathbf{0.9505 \pm 0.0035}$ |
| C | DiffuSETS | $0.1092 \pm 0.0022$ | $13.29 \pm 1.13$ | $0.9242 \pm 0.0039$ | $0.9209 \pm 0.0038$ |
| C | **SE-Diff** | $\mathbf{0.0923 \pm 0.0021}$ | $\mathbf{8.43 \pm 0.42}$ | $\mathbf{0.9393 \pm 0.0031}$ | $\mathbf{0.9509 \pm 0.0035}$ |

## J    SENSITIVITY ANALYSIS OF EXPERIENCE RETRIEVAL–AUGMENTED CONDITIONING

Table 10 summarizes the sensitivity of the retrieval performance to the tri-view weights $(\lambda_1, \lambda_2, \lambda_3)$ (diagnoses / prescriptions / procedures) and the number of retrieved admissions $k$. With equal weights $(\lambda_1, \lambda_2, \lambda_3) = (1, 1, 1)$, varying $k \in 2, 3, 4$ yields very similar mean scores (94.4–94.8), with a slight peak at $k = 3$ ($94.9 \pm 2.3$), indicating limited sensitivity to $k$ in this range. For $k = 3$, varying one weight at a time while keeping the others at 1 shows that upweighting any single view changes performance by at most $\sim 1.5$ points, suggesting that the tri-view retrieval is reasonably robust and that the symmetric setting $(\lambda_1, \lambda_2, \lambda_3) = (1, 1, 1)$ with $k = 3$ is a simple and competitive

default used in all main experiments. Retrieval performance is quantified by asking an LLM (GPT-5) to score each retrieved experience report on a 1–100 scale and reporting the mean $\pm$ standard deviation of these scores.

Table 10: Sensitivity of tri-view weights $(\lambda_1, \lambda_2, \lambda_3)$ and top-$k$ on retrieval performance.

| $\lambda_1$ | $\lambda_2$ | $\lambda_3$ | $k$ | LLM Score |
|---|---|---|---|---|
| 1 | 1 | 1 | 3 | **94.9 $\pm$ 2.3** |
| 1 | 1 | 1 | 2 | 94.4 $\pm$ 2.1 |
| 1 | 1 | 1 | 4 | 94.8 $\pm$ 2.9 |
| 2 | 1 | 1 | 3 | 94.7 $\pm$ 2.6 |
| 1 | 2 | 1 | 3 | 93.6 $\pm$ 1.5 |
| 1 | 1 | 2 | 3 | 93.5 $\pm$ 3.3 |
| 3 | 1 | 1 | 3 | 94.3 $\pm$ 2.0 |
| 1 | 3 | 1 | 3 | 94.6 $\pm$ 1.6 |
| 1 | 1 | 3 | 3 | 94.2 $\pm$ 1.4 |

## K EXAMPLE OF EXPERIENCE RETRIEVAL–AUGMENTED CONDITIONING

To make the experience retrieval–augmented conditioning pipeline more concrete, we provide a representative example below.

- **Original machine-generated diagnosis and metadata.**
  Original machine-generated diagnosis: *Sinus rhythm*.
  Heart rate (bpm): 83.
  Age (years): 91.
  Gender: F.

- **Top-$k$ retrieved similar admissions (diagnosis view, $k = 3$).**
  *Sinus rhythm*
  *Sinus rhythm*
  *Sinus rhythm with borderline 1st degree A–V block*

- **LLM-generated experience report.**
  The ECG demonstrates a sinus rhythm at a rate of 83 beats per minute in a 91-year-old female. The rhythm is regular with normal P wave morphology preceding each QRS complex, consistent with sinus node origin. There is no evidence of atrioventricular conduction delay or block; PR interval is within normal limits. QRS complexes are narrow, indicating normal ventricular conduction. No arrhythmias or ectopic beats are noted. Overall, the tracing reflects a stable sinus rhythm appropriate for the patient's age without signs of sinus tachycardia or conduction abnormalities.

