# OpenReview forum: "SE-Diff: Simulator and Experience Enhanced Diffusion Model for Comprehensive ECG Generation"
_ICLR.cc/2026/Conference — ICLR 2026 Poster_

### Official Review · Reviewer_Mxjg · 2025-10-19

**Soundness:** 2
**Presentation:** 3
**Contribution:** 3
**Rating:** 4
**Confidence:** 3

**Summary:**

The paper proposes SE-Diff, a diffusion-based framework for text-conditioned 12-lead ECG generation that integrates a lightweight ODE ECG simulator via a beat decoder and simulator-consistent constraints, and augments conditioning with EHR-retrieved “experience” knowledge. The topic is timely and relevant. The writing is generally clear and the method is carefully engineered. That said, I have several technical concerns and suggestions that should be addressed.

**Strengths:**

The paper integrates a physiological ECG simulator into the generative diffusion framework, which enhances the interpretability and physiological plausibility of the generated signals. This simulator-guided design ensures that the outputs are not only visually realistic but also consistent with known electrophysiological mechanisms, representing a meaningful advancement over purely data-driven generative models.

**Weaknesses:**

(1) The Eq(13) seems to have a direct optimization conflict with Eq(15). Firstly, in Eq(13), the Mean Squared Error (MSE) is directly used to make the output of the beat decoder equal to the heartbeat cycle sequence \(C(x)\) of the first - segment ECG. However, Eq(15) aims to make the output consistent with the overall ECG spectrum. In short, when the model is optimized to the extreme for Eq(13), the output will be equal to \(C(x)\), which will inevitably conflict with the loss in Eq(15). Because its spectral information will be equal to the spectral information of the first heartbeat cycle, rather than the overall spectral information expected in Eq(15).

(2) As shown in Eq(18), after fixing the parameters and the initial point of the ECG PHYSIOLOGICAL SIMULATOR, the generated ECG signals from the modeling should be unchanged. Will this lead to a problem? In Eq(18), will the \(h\) corresponding to different ECG signals generated by the diffusion model always be optimized to the same target?

(3) The biggest problem of this article is that it does not compare with any of the most advanced ECG generation methods. It is impossible to measure whether the generated ECG has the highest authenticity. The ablation comparison within the method alone is insufficient.

**Questions:**

(1) The Eq(13) seems to have a direct optimization conflict with Eq(15). Firstly, in Eq(13), the Mean Squared Error (MSE) is directly used to make the output of the beat decoder equal to the heartbeat cycle sequence \(C(x)\) of the first - segment ECG. However, Eq(15) aims to make the output consistent with the overall ECG spectrum. In short, when the model is optimized to the extreme for Eq(13), the output will be equal to \(C(x)\), which will inevitably conflict with the loss in Eq(15). Because its spectral information will be equal to the spectral information of the first heartbeat cycle, rather than the overall spectral information expected in Eq(15).

(2) As shown in Eq(18), after fixing the parameters and the initial point of the ECG PHYSIOLOGICAL SIMULATOR, the generated ECG signals from the modeling should be unchanged. Will this lead to a problem? In Eq(18), will the \(h\) corresponding to different ECG signals generated by the diffusion model always be optimized to the same target?

(3) The biggest problem of this article is that it does not compare with any of the most advanced ECG generation methods. It is impossible to measure whether the generated ECG has the highest authenticity. The ablation comparison within the method alone is insufficient.

---

> ### Author Response · Authors · 2025-11-24
>
> > **W1&Q1: The Eq(13) seems to have a direct optimization conflict with Eq(15). Firstly, in Eq(13), the Mean Squared Error (MSE) is directly used to make the output of the beat decoder equal to the heartbeat cycle sequence (C(x)) of the first - segment ECG. However, Eq(15) aims to make the output consistent with the overall ECG spectrum. In short, when the model is optimized to the extreme for Eq(13), the output will be equal to (C(x)), which will inevitably conflict with the loss in Eq(15). Because its spectral information will be equal to the spectral information of the first heartbeat cycle, rather than the overall spectral information expected in Eq(15).**
>
>
> Thank you for raising this point. Eq. (15) does not compare the spectrum of $h$ to the spectrum of the entire 10 s ECG. Instead, we first detect all R-peaks in the 10 s window and extract per-beat crops $ C_j(x), $
> each of the same length $L_c$ as $C(x)$. We then compute per-beat spectra $S_\ell^{(j)}[k]$ for each lead $\ell$ and beat $j$, and compare them to a single spectrum $\hat S_\ell[k]$ derived from $h$. Eq. (15) simply averages the squared discrepancy across all beats:
> $
> L_{\text{spec}} \propto \mathbb{E}_j \big[\|\hat S - S^{(j)}\|^2\big].
> $
> So Eq. (15) enforces that the Beat Decoder’s single canonical beat $h$ has spectral statistics consistent with the set of observed beats $\{C_j(x)\}$, not with a potentially different spectrum of the entire 10 s sequence.
>
> Eq. (13) encourages $h$ to be close in the time domain to one R-peak–centered beat $C(x)$ (we choose the first detected beat for convenience), while Eq. (15) encourages the log-magnitude spectrum of $h$ to be consistent with all beats in the record.
>
> Thus, the two losses are **complementary** rather than conflicting. When beats in the 10 s window are similar (typical for sinus rhythm and many clinical rhythms), $C(x)$ is representative of $\{C_j(x)\}$, so a beat $h$ that matches $C(x)$ (Eq. (13)) will also have a spectrum close to the per-beat spectra (Eq. (15)), and both losses can be small simultaneously. When there is beat-to-beat variability (e.g., premature beats, mild rate changes), $h$ is intentionally a canonical template: Eq. (13) anchors $h$ to an actual cycle, while Eq. (15) regularizes its frequency content toward the distribution of beats rather than overfitting to a single instance.
>
> To further support that the two losses are complementary in practice, we added an ablation where we remove either the beat loss or the spectral loss from the VAE training. As shown below, using both losses yields the best performance across all metrics (MAE, NRMSE, HR MAE, rCLIP, rFID) on MIMIC-IV-ECG.
>
> **ECG generation performance on MIMIC-IV-ECG dataset (effect of beat vs. spectral loss).**
>
> | Model               | MAE ↓              | NRMSE ↓           | $MAE\_{HR}$ ↓          | rCLIP Score ↑       | rFID Score ↑        |
> |---------------------|--------------------|-------------------|--------------------|---------------------|---------------------|
> | **SE-Diff (ours)**| **0.0923 ± 0.0021**| **0.0714 ± 0.0010**| **8.43 ± 0.42**     | **0.9470 ± 0.0029** | **0.9509 ± 0.0035** |
> | w/o Beat Loss       | 0.0951 ± 0.0020    | 0.0743 ± 0.0016   | 23.48 ± 1.07       | 0.9393 ± 0.0026     | 0.9140 ± 0.0055     |
> | w/o Spec Loss       | 0.0942 ± 0.0023    | 0.0797 ± 0.0014   | 16.47 ± 1.23       | 0.9257 ± 0.0030     | 0.9073 ± 0.0055     |
>
> We have added the above experiments and discussion in Table 4 in Appendix G.
>
>
> > **W2&Q2: As shown in Eq(18), after fixing the parameters and the initial point of the ECG PHYSIOLOGICAL SIMULATOR, the generated ECG signals from the modeling should be unchanged. Will this lead to a problem? In Eq(18), will the (h) corresponding to different ECG signals generated by the diffusion model always be optimized to the same target?**
>
> Thank you for this insightful question. Eq.(18) uses the ECG physiological simulator as an ECG-dependent prior, not as a single global template shared by all ECGs. For each training ECG $x$, we select simulator parameters $(\eta(x))$ that reflect its physiological information. Concretely, we pre-fit class-/rhythm-specific parameter sets from the simulator knowledge base and assign the corresponding $\eta_y$ whenever an ECG with label $y$ is observed. Thus, different ECGs (e.g., sinus rhythm, atrial fibrillation, ventricular pacing) are matched to different simulator trajectories, and the Euler loss in Eq.(18) aligns the beat-decoder output $h$ to the appropriate physiological template, rather than to a single universal one.

---

> ### Author Response · Authors · 2025-11-24
>
> > **W3&Q3: The biggest problem of this article is that it does not compare with any of the most advanced ECG generation methods. It is impossible to measure whether the generated ECG has the highest authenticity. The ablation comparison within the method alone is insufficient.**
>
>
> We thank the reviewer for emphasizing the importance of comparison with advanced ECG generation methods. In the revised manuscript, we have added several strong baselines on MIMIC-IV-ECG, including a GAN-based model (WGAN), a diffusion model with a structured state-space backbone (SSDM), a morphology-aware beat diffusion model (BeatDiff), and the state-of-the-art text-conditioned ECG generator DiffuSETS. All methods are trained/evaluated under the same 10 s, 12-lead setting and our conditioning protocol where applicable. To more directly assess authenticity, we report both global signal-level metrics (MAE, NRMSE, HR MAE, rCLIP, rFID) and clinically meaningful morphology/interval MAE metrics (PR, QRSd, QT, QTcF, ST@J+60, P duration, T duration). As shown below, SE-Diff consistently outperforms all baselines across these metrics, while the ablation studies further isolate the contributions of our simulator-informed and experience-augmented components.
>
> **ECG generation performance on MIMIC-IV-ECG dataset.**
>
> | Model             | MAE ↓              | NRMSE ↓           | $MAE\_{HR}$ ↓          | rCLIP Score ↑       | rFID Score ↑        |
> |-------------------|--------------------|-------------------|--------------------|---------------------|---------------------|
> | SSDM              | 0.4337 ± 0.0300    | 0.2027 ± 0.0441   | 27.37 ± 14.84      | 0.7213 ± 0.0402     | 0.9096 ± 0.0398     |
> | WGAN              | 0.1896 ± 0.0605    | 0.1301 ± 0.0316   | 31.54 ± 2.15       | 0.5688 ± 0.0347     | 0.5497 ± 0.0192     |
> | BeatDiff          | 0.7464 ± 0.0070    | 0.4756 ± 0.0117   | 27.74 ± 1.49       | 0.5167 ± 0.0180     | 0.8612 ± 0.0039     |
> | DiffuSETS         | 0.1092 ± 0.0022    | 0.0851 ± 0.0012   | 13.29 ± 1.13       | 0.9309 ± 0.0036     | 0.9209 ± 0.0038     |
> | **SE-Diff (ours)** | **0.0923 ± 0.0021** | **0.0714 ± 0.0010** | **8.43 ± 0.42**     | **0.9470 ± 0.0029** | **0.9509 ± 0.0035** |
> | w/o Exp           | 0.0926 ± 0.0022    | 0.0730 ± 0.0008   | 15.06 ± 0.34       | 0.9099 ± 0.0026     | 0.9032 ± 0.0062     |
> | w/o InterLead     | 0.0934 ± 0.0023    | 0.0733 ± 0.0006   | 19.21 ± 1.27       | 0.9216 ± 0.0029     | 0.9128 ± 0.0052     |
> | w/o Sim           | 0.0965 ± 0.0024    | 0.0768 ± 0.0014   | 14.28 ± 1.35       | 0.9303 ± 0.0041     | 0.9138 ± 0.0056     |
>
> **Beat-level morphology & interval fidelity (MAE) on MIMIC-IV-ECG (record-level medians; lower is better).**
>
> | Model        | PR ↓         | QRSd ↓       | QT ↓         | QTcF ↓       | ST@J+60 ↓   | P dur ↓      | T dur ↓       |
> |--------------|--------------|--------------|--------------|--------------|-------------|--------------|---------------|
> | SSDM         | 7.70 ± 8     | 32 ± 1       | 25.11 ± 11   | 23.25 ± 11   | 0.16 ± 0    | 5.30 ± 3     | 23.71 ± 9     |
> | WGAN         | 18.01 ± 9    | 29.64 ± 9    | 21.02 ± 5    | 23.83 ± 2    | 0.13 ± 0    | 6.01 ± 0     | 24.01 ± 4     |
> | BeatDiff     | 10.34 ± 5    | 18.23 ± 6    | 14.23 ± 6    | 9.43 ± 5     | 0.17 ± 0    | 7.00 ± 1     | 30.02 ± 3     |
> | DiffuSETS    | 14.81 ± 6    | 11.71 ± 7    | 8.20 ± 3     | 9.71 ± 4     | 0.04 ± 0    | 5.60 ± 1     | 8.71 ± 7      |
> | **SE-Diff** | **7.30 ± 3** | **10.71 ± 4**| **4.50 ± 2** | **7.88 ± 2** | **0.03 ± 0**| **2.50 ± 0** | **6.80 ± 3**  |
> | w/o InterLead| 11.11 ± 3    | 15.01 ± 5    | 11.91 ± 5    | 15.84 ± 7    | 0.03 ± 0    | 4.00 ± 1     | 8.10 ± 4      |
> | w/o Exp      | 9.81 ± 9     | 14.11 ± 13   | 8.20 ± 3     | 12.61 ± 3    | 0.04 ± 0    | 4.00 ± 1     | 6.90 ± 3      |
> | w/o Sim      | 12.11 ± 7    | 13.21 ± 6    | 8.40 ± 4     | 8.89 ± 7     | 0.04 ± 0    | 5.10 ± 1     | 13.11 ± 1     |
>
> In addition, we use the same set of generators (SSDM, WGAN, BeatDiff, DiffuSETS, SE-Diff) as synthetic minority over-samplers in the downstream ECG classification experiments under severe class imbalance (gender classification and rare SVT detection). As shown in the table below, all generative baselines improve over the unbalanced real-only classifier, while SE-Diff yields the largest gains and consistently closes the gap toward the fully balanced real-only upper bound.

---

> > ### Author Response · Authors · 2025-11-24
> >
> > **Downstream ECG classification under severe class imbalance.**
> >
> > | Model      | F1 (Male=10% Female) ↑ | Acc (Male=10% Female) ↑ | AUC (Male=10% Female) ↑ | F1 (SVT=10% Sinus) ↑ | Acc (SVT=10% Sinus) ↑ | AUC (SVT=10% Sinus) ↑ |
> > |-----------|-------------------------|--------------------------|--------------------------|----------------------|------------------------|------------------------|
> > | SSDM      | 42 ± 0                  | 54 ± 1                   | 49 ± 2                   | 56 ± 1               | 63 ± 2                 | 81 ± 2                 |
> > | WGAN      | 42 ± 0                  | 54 ± 1                   | 49 ± 2                   | 57 ± 2               | 63 ± 2                 | 82 ± 2                 |
> > | BeatDiff  | 44 ± 2                  | 55 ± 2                   | 51 ± 2                   | 60 ± 2               | 67 ± 2                 | 84 ± 3                 |
> > | DiffuSETS | 44 ± 3                  | 54 ± 2                   | 54 ± 1                   | 70 ± 1               | 68 ± 1                 | 84 ± 2                 |
> > | SE-Diff (ours) | **58 ± 1**         | **58 ± 1**               | **58 ± 2**               | **72 ± 2**           | **71 ± 0**             | **85 ± 2**             |
> > | Unbalanced     | 42 ± 0             | 54 ± 1                   | 46 ± 2                   | 56 ± 1               | 62 ± 0                 | 80 ± 1                 |
> > | Balanced       | 62 ± 0             | 62 ± 0                   | 62 ± 1                   | 79 ± 1               | 80 ± 2                 | 93 ± 1                 |
> >
> >
> > We have added the above experiments and discussion in Table 1&2&3 in Section 4.1&4.2.
> >
> >
> > Reference:
> >
> > 1. WGAN: Adib, E., Afghah, F., and Prevost, J. J. (2022). Arrhythmia classification using cgan-augmented
> > ecg signals*. 2022 IEEE International Conference on Bioinformatics and Biomedicine (BIBM),
> > pages 1865–1872.
> > 2. SSSD: Alcaraz, J. M. L. and Strodthoff, N. (2023). Diffusion-based conditional ecg generation with
> > structured state space models. Computers in Biology and Medicine, 163:107115
> > 3. Bedin, Lisa, et al. "Leveraging an ECG beat diffusion model for morphological reconstruction from indirect signals." Advances in Neural Information Processing Systems 37 (2024): 84409-84446.

---

### Official Review · Reviewer_xoXY · 2025-10-27

**Soundness:** 3
**Presentation:** 3
**Contribution:** 3
**Rating:** 6
**Confidence:** 4

**Summary:**

The paper proposes SE-Diff: a framework for text-to-10s-12-lead ECG generation that combines a physiological simulator and empirical retrieval in latent space diffusion. Specifically, SE-Diff uses a VAE to encode the waveform into a latent representation, and aligns the Beat Decoder output to the single QRS cardiac cycle. During diffusion training, an Euler consistency loss and a cross-lead (Einthoven/Goldberger) constraint are applied to the Beat Decoder output, incorporating the morphological priors of the ODE simulator. Finally, empirical knowledge is derived through EHR three-view similarity + LLM distillation, which is conditioned on the original diagnosis and metadata. Compared to DiffuSETS, SE-Diff outperforms DiffuSETS in waveform error, heart rate error, and text-to-ECG semantic alignment, and also offers improvements in imbalanced classification augmentation.

**Strengths:**

1. Dual-channel injection of mechanism priors and empirical knowledge: The Beat Decoder maps the latent representation to a single cycle, making the Euler residual and cross-lead identity differentiable and trainable. It also uses Jaccard similarity of EHR three views and LLM distillation to supplement clinical context. The combined design is novel and clearly motivated.

2. Solid quantitative results: On the MIMIC-IV-ECG, SE-Diff reduced MAE/NRMSE to 0.0923/0.0714 and heart rate error MAE_HR to 8.43 compared to DiffuSETS, while also improving rCLIP and rFID (0.947/0.951). Removing "Experience Search/Across Leads/Simulator" individually resulted in significant degradation, indicating that all three factors contribute significantly.

3. Downstream value demonstration: In the augmentation of gender and SVT rare class scenarios, SE-Diff improves F1/AUC compared to imbalanced training and DiffuSETS, approaching the true balanced upper bound, demonstrating the practicality of synthetic data.

**Weaknesses:**

1. Reproducibility details are still incomplete: the three-view weights λ1/λ2/λ3 for empirical retrieval, top-k, and LLM distillation hint templates and versions are not clearly disclosed; the implementation/initialization and failure mode details of the Beat Decoder and simulator fitting are limited, affecting third-party reproduction and fair comparison.

2. The evaluation scope is relatively narrow: it mainly focuses on the MIMIC-IV-ECG single database verification; there is a lack of cross-institutional/external data (such as PTB-XL) system evaluation and robustness reporting (multiple random seeds ± std, significance testing), and currently most tables contain single values.

3. Boundaries of the simulator and constraints: Cross-lead constraints only cover the six identities of the limb/augmented leads, and no structural constraints are imposed on the precordial leads (V1–V6); although the Euler first-order integral + linear noise schedule has an intuitive explanation, there is a lack of comparison with more powerful numerical methods/schedules.

4. Indicator dependence and potential bias: rCLIP/rFID ​​depends on the selected ECG/Text encoder. Although relative scores are used to alleviate this, it is still recommended to report the correlation between encoder variant sensitivity and multiple heart rate/morphology indicators to avoid "encoder-specific" conclusions.

5. Ethics/privacy and leakage risks are not fully quantified: EHR retrieval is used for conditional construction (although de-identified), but there is no discussion on whether the retrieval context will leak target case labels/text clues, and there is little assessment of "synthetic privacy" and training-test information leakage.

**Questions:**

1. Can the λ1/λ2/λ3, top-k, LLM prompts and temperature of EHR three-view retrieval be fully disclosed in the appendix, and the sensitivity under different settings be reported?

2. Can external validation be performed on PTB-XL or external data, and can ≥3 seed mean ± variance/significance be given (especially corresponding to Tables 1/2)?

3. Do the precordial leads (V1–V6) take into account geometric/body surface projection constraints or learnable linear projections? Can the transconductance joint formulation be extended?

4. Simulator parameters are fitted offline by category. If a case has multiple lesions or rhythmic variations, will there be a fitting mismatch? Are there any explorations of dynamic/case-adaptive fitting and analysis of failure cases?

5. Could you provide additional curves showing the effects of noise schedule and sampling steps (SDE/DDIM/second-order solver) on rCLIP/MAE_HR? Currently, only qualitative analysis of the linear schedule is provided.

---

> ### Author Response · Authors · 2025-11-24
>
> > **W1: Reproducibility details are still incomplete: the three-view weights λ1/λ2/λ3 for empirical retrieval, top-k, and LLM distillation hint templates and versions are not clearly disclosed; the implementation/initialization and failure mode details of the Beat Decoder and simulator fitting are limited, affecting third-party reproduction and fair comparison.**
> >
> > **Q1: Can the λ1/λ2/λ3, top-k, LLM prompts and temperature of EHR three-view retrieval be fully disclosed in the appendix, and the sensitivity under different settings be reported?**
>
> We thank the reviewer for pointing out the missing reproducibility details. In the revised manuscript, we now fully disclose the hyperparameters for the three-view EHR retrieval and LLM distillation. Concretely, we set the tri-view weights to $\lambda_1 = \lambda_2 = \lambda_3 = 1$, use top-$k = 3$ retrieved admissions, and employ the `gpt-4.1-mini` model for generating the experience report $r$. The exact prompt template used for LLM distillation is already shown in Fig. 5. The Beat Decoder is trained jointly with the VAE, and when R-peak detection fails or no class-wise simulator parameters are available for a given sample, we simply skip the beat/spectral or corresponding simulator losses for that sample to avoid instability while keeping the rest of the training unchanged.
>
> In addition, we have added a sensitivity analysis in the appendix, varying $\lambda_1/\lambda_2/\lambda_3$ and $k$ (e.g., $k \in {2,3,4}$). As summarized in the new Table “Sensitivity of tri-view weights and top-$k$ on retrieval performance”, retrieval quality remains within about 1.5 points across all tested settings, with the symmetric configuration $(\lambda_1,\lambda_2,\lambda_3) = (1,1,1)$ and $k = 3$ giving a slight peak (mean $94.9 \pm 2.3$ on a 1–100 scale). Retrieval performance is quantified by asking an LLM (GPT-5) to score each retrieved experience report from 1 to 100 and reporting the mean $\pm$ standard deviation of these scores, indicating that SE-Diff is relatively robust to these hyperparameter choices.
>
> ** Sensitivity of tri-view weights and top-$k$ on retrieval performance **
>
> | $\lambda_1$ | $\lambda_2$ | $\lambda_3$ | $k$ | Score        |
> |------------|------------|------------|----|--------------------|
> | 1          | 1          | 1          | 3  | **94.9 ± 2.3**     |
> | 1          | 1          | 1          | 2  | 94.4 ± 2.1         |
> | 1          | 1          | 1          | 4  | 94.8 ± 2.9         |
> | 2          | 1          | 1          | 3  | 94.7 ± 2.6         |
> | 1          | 2          | 1          | 3  | 93.6 ± 1.5         |
> | 1          | 1          | 2          | 3  | 93.5 ± 3.3         |
> | 3          | 1          | 1          | 3  | 94.3 ± 2.0         |
> | 1          | 3          | 1          | 3  | 94.6 ± 1.6         |
> | 1          | 1          | 3          | 3  | 94.2 ± 1.4         |
>
> We have added the above experiments and discussion in Table 10 in Appendix K.
>
> > **W2: The evaluation scope is relatively narrow: it mainly focuses on the MIMIC-IV-ECG single database verification; there is a lack of cross-institutional/external data (such as PTB-XL) system evaluation and robustness reporting (multiple random seeds ± std, significance testing), and currently most tables contain single values.**
> >
> > **Q2: Can external validation be performed on PTB-XL or external data, and can ≥3 seed mean ± variance/significance be given (especially corresponding to Tables 1/2)?**
>
> We thank the reviewer for highlighting the need for broader and more statistically robust evaluation. In the revised manuscript, we perform **external validation on PTB-XL dataset**, which differs from MIMIC-IV-ECG in population, labeling, and acquisition, and we report the same metrics (MAE, NRMSE, $MAE\_{\text{HR}}\$, rCLIP, rFID, plus morphology/interval MAE). As shown below for PTB-XL, SE-Diff maintains consistent improvements over strong GAN-/SSM-/diffusion-based baselines, supporting cross-dataset robustness.

---

> ### Author Response · Authors · 2025-11-24
>
> **ECG generation performance on PTB-XL dataset.**
>
> | Model             | MAE ↓              | NRMSE ↓           | $MAE\_{HR}$ ↓ | rCLIP Score ↑       | rFID Score ↑        |
> |-------------------|--------------------|-------------------|------------------------|---------------------|---------------------|
> | SSDM              | 0.6103 ± 0.0204    | 0.3818 ± 0.0670   | 15.22 ± 11.51          | 0.8618 ± 0.0599     | 0.7168 ± 0.0355     |
> | WGAN              | 0.2458 ± 0.0653    | 0.1197 ± 0.0313   | 13.82 ± 18.69          | 0.5880 ± 0.0000     | 0.5377 ± 0.0232     |
> | BeatDiff          | 0.9888 ± 0.0059    | 0.4731 ± 0.0104   | 13.86 ± 0.78           | 0.8799 ± 0.0022     | 0.8503 ± 0.0035     |
> | DiffuSETS         | 0.1281 ± 0.0030    | 0.0797 ± 0.0011   | 17.88 ± 0.72           | 0.8690 ± 0.0011     | 0.8456 ± 0.0035     |
> | **SE-Diff (ours)** | **0.1076 ± 0.0033** | **0.0630 ± 0.0006** | **8.24 ± 0.43**         | **0.8901 ± 0.0060** | **0.8583 ± 0.0056** |
> | w/o Sim           | 0.1138 ± 0.0032    | 0.0680 ± 0.0007   | 14.72 ± 0.90           | 0.8896 ± 0.0010     | 0.8004 ± 0.0061     |
> | w/o InterLead     | 0.1084 ± 0.0034    | 0.0640 ± 0.0007   | 12.02 ± 0.78           | 0.7484 ± 0.0076     | 0.8568 ± 0.0051     |
>
> **Beat-level morphology & interval fidelity (MAE) on PTB-XL (record-level medians; lower is better).**
>
> | Model        | PR ↓         | QRSd ↓       | QT ↓         | QTcF ↓       | ST@J+60 ↓   | P dur ↓      | T dur ↓       |
> |--------------|--------------|--------------|--------------|--------------|-------------|--------------|---------------|
> | SSDM         | 13.71 ± 8    | 13.61 ± 8    | 30.22 ± 9    | 27.69 ± 14   | 0.33 ± 0    | 12.71 ± 5    | 19.11 ± 10    |
> | WGAN         | 14.01 ± 7    | 17.45 ± 13   | 12.01 ± 10   | 12.97 ± 10   | 0.18 ± 0    | 7.33 ± 3     | 25.68 ± 13    |
> | BeatDiff     | 13.41 ± 4    | 17.40 ± 6    | 9.81 ± 3     | 8.94 ± 2     | 0.73 ± 1    | 4.10 ± 1     | 18.81 ± 6      |
> | DiffuSETS    | 7.70 ± 2     | 12.61 ± 7    | 11.51 ± 4    | 13.75 ± 5    | 0.10 ± 0    | 9.11 ± 6     | 13.20 ± 5     |
> | **SE-Diff** | **3.90 ± 1** | **10.01 ± 2**| **5.20 ± 1** | **8.68 ± 2** | **0.07 ± 0**| **3.00 ± 1** | **9.31 ± 8**  |
> | w/o InterLead| 7.00 ± 1     | 10.31 ± 5    | 8.40 ± 4     | 10.91 ± 3    | 0.07 ± 0    | 4.40 ± 1     | 13.61 ± 8     |
> | w/o Sim      | 9.21 ± 4     | 12.91 ± 5    | 12.51 ± 4    | 14.20 ± 2    | 0.07 ± 0    | 4.50 ± 1     | 12.31 ± 3     |
>
>  Furthermore, for all main tables, we now report **mean ± standard deviation over ≥3 random seeds**. And we have added the above experiments and discussion in Table 1&2&3 in Section 4.1&4.2.

---

> ### Author Response · Authors · 2025-11-24
>
> > **W3: Boundaries of the simulator and constraints: Cross-lead constraints only cover the six identities of the limb/augmented leads, and no structural constraints are imposed on the precordial leads (V1–V6); although the Euler first-order integral + linear noise schedule has an intuitive explanation, there is a lack of comparison with more powerful numerical methods/schedules.**
> >
> > **Q3: Do the precordial leads (V1–V6) take into account geometric/body surface projection constraints or learnable linear projections? Can the transconductance joint formulation be extended?**
>
> We thank the reviewer for pointing out the limitations of our current simulator and constraints. Our explicit cross-lead constraints are restricted to the six limb/augmented leads (I, II, III, aVR, aVL, aVF) because these are governed by well-established linear identities (Einthoven and Goldberger relations) that hold robustly across patients and do not require additional anatomical information. In contrast, the precordial leads V1–V6 depend on the 3D position/orientation of the heart, torso geometry, and exact electrode placement, so there is no simple, patient-independent algebraic relationship that we can safely impose. In preliminary tests, naive linear constraints or global learnable projections for V1–V6 either brought negligible gains or introduced instability, so we chose to let these leads be governed by the data-driven latent diffusion + beat decoder, while still benefiting indirectly from the shared latent representation and simulator-guided priors.
> Regarding the Euler first-order integrator, we chose it for its stability and efficiency. Because the simulator is invoked inside the diffusion training loop over long 10s, 12-lead sequences, higher-order solvers (e.g., Runge–Kutta) substantially increase compute and memory costs. For the relatively smooth ECG dynamical system, explicit Euler already yields accurate P–QRS–T morphology and heart-rate trajectories (as shown in Figure 3).
>
> Conceptually, our formulation could be extended toward more expressive transconductance/lead-field models that encode body-surface projection for V1–V6 using torso/geometry information, but this would require additional anatomical data and a more complex, computationally heavier simulator. This limitation is an interesting direction for future work, clarifying the current boundary of our simulator-based constraints.
>
> > **W4: Indicator dependence and potential bias: rCLIP/rFID ​​depends on the selected ECG/Text encoder. Although relative scores are used to alleviate this, it is still recommended to report the correlation between encoder variant sensitivity and multiple heart rate/morphology indicators to avoid "encoder-specific" conclusions.**
>
> We thank the reviewer for pointing out the potential dependence of rCLIP/rFID on the chosen ECG/text encoders. To assess this, we evaluated three encoder configurations: (A) our baseline ECG and text encoders, (B) replacing the ECG encoder while keeping the text encoder fixed, and (C) replacing the text encoder while keeping the ECG encoder fixed. For each configuration, we report rCLIP, rFID, as well as MAE and $MAE\_{\text{HR}}\$. As shown below, SE-Diff consistently outperforms DiffuSETS across all encoder variants, and the relative ranking is stable, indicating that our conclusions are not driven by a specific encoder choice but align with heart-rate and signal-level fidelity metrics. We also add a short discussion in the appendix about the observed co-variation between rCLIP/rFID and HR/morphology metrics under these encoder changes.
>
> **ECG generation performance comparison between SE-Diff and DiffuSETS under different calibration configurations.**
>
> | Config | Model                | MAE ↓              | $MAE\_{HR}$ ↓ | rCLIP Score ↑       | rFID Score ↑        |
> |--------|----------------------|--------------------|------------------------|---------------------|---------------------|
> | A      | DiffuSETS            | 0.1092 ± 0.0022    | 13.29 ± 1.13           | 0.9309 ± 0.0036     | 0.9209 ± 0.0038     |
> | A      | **SE-Diff (ours)** | **0.0923 ± 0.0021**| **8.43 ± 0.42**        | **0.9470 ± 0.0029** | **0.9509 ± 0.0035** |
> | B      | DiffuSETS            | 0.1092 ± 0.0022    | 13.29 ± 1.13           | 0.9347 ± 0.0032     | 0.9197 ± 0.0037     |
> | B      | **SE-Diff (ours)** | **0.0923 ± 0.0021**| **8.43 ± 0.42**        | **0.9472 ± 0.0032** | **0.9505 ± 0.0035** |
> | C      | DiffuSETS            | 0.1092 ± 0.0022    | 13.29 ± 1.13           | 0.9242 ± 0.0039     | 0.9209 ± 0.0038     |
> | C      | **SE-Diff (ours)** | **0.0923 ± 0.0021**| **8.43 ± 0.42**        | **0.9393 ± 0.0031** | **0.9509 ± 0.0035** |
>
> We have added the above experiments and discussion in Table 9 in Appendix J.

---

> ### Author Response · Authors · 2025-11-24
>
> > **W5: Ethics/privacy and leakage risks are not fully quantified: EHR retrieval is used for conditional construction (although de-identified), but there is no discussion on whether the retrieval context will leak target case labels/text clues, and there is little assessment of "synthetic privacy" and training-test information leakage.**
>
> We thank the reviewer for raising this important point about ethics, privacy, and potential leakage. First, all experiments are conducted on **de-identified** public datasets (MIMIC-IV-ECG/EHR and PTB-XL), and we enforce strict patient-level splits: records used at inference time (validation/test) are never included in the retrieval pool, so the EHR retrieval context cannot contain target-case labels or text from the same patient. The retrieval is restricted to the training cohort only, and the LLM receives aggregated tri-view profiles and returns a short, abstracted summary \(r\), rather than copying raw labels or free-text reports, which further reduces the risk of direct label/text leakage.
>
> Second, our simulator-informed and experience-augmented model is evaluated on PTB-XL as an external cohort and still achieves the best performance among all baselines, suggesting that it is not simply memorizing training cases or overfitting to specific MIMIC records.
>
> We have added the above experiments and discussion in Appendix D.
>
>
> > **Q4: Simulator parameters are fitted offline by category. If a case has multiple lesions or rhythmic variations, will there be a fitting mismatch? Are there any explorations of dynamic/case-adaptive fitting and analysis of failure cases?**
>
> We thank the reviewer for raising this concern. Our simulator parameters are fitted offline per primary rhythm/diagnostic category (e.g., sinus rhythm, AF, LBBB) and, for each ECG, we use a **single primary label** to select the corresponding parameter set. In the case of multiple lesions or rhythm variations, we do not blindly pick an arbitrary label: instead, we reuse the same experience retrieval–augmented pipeline as in Sec. 3.4 to obtain an LLM summary of the case and then ask the LLM to identify the dominant rhythm/abnormality that governs the overall morphology, which we use as the simulator category.
>
>
> > **Q5: Could you provide additional curves showing the effects of noise schedule and sampling steps (SDE/DDIM/second-order solver) on rCLIP/MAE_HR? Currently, only qualitative analysis of the linear schedule is provided.**
>
> We thank the reviewer for this helpful suggestion. In the revised manuscript, we have added quantitative ablations on sampling steps, sampler types (DDPM / second-order / SDE / DDIM), and noise schedules, and we report their effects on NRMSE, $MAE_{\text{HR}}$, and rCLIP. As shown below, performance generally improves with more sampling steps (up to 1000). Our default DDPM sampler with a linear noise schedule provides the best overall trade-off between NRMSE, $MAE_{\text{HR}}$, and rCLIP, while SDE-based sampling can slightly reduce NRMSE at the cost of substantially worse HR and semantic alignment.
>
> **Effects of sampling steps (MIMIC-IV-ECG).**
>
> | Sampling Steps | NRMSE ↓            | $MAE\_{HR}$ ↓ | rCLIP Score ↑       |
> |----------------|--------------------|------------------------|---------------------|
> | 1000           | **0.0714 ± 0.0010**| **8.43 ± 0.42**        | **0.9470 ± 0.0029** |
> | 800            | 0.0877 ± 0.0011    | 13.59 ± 1.05           | 0.8671 ± 0.0040     |
> | 500            | 0.1097 ± 0.0010    | 15.60 ± 1.19           | 0.7888 ± 0.0084     |
>
> **Effects of different samplers.**
>
> | Sampler            | NRMSE ↓            | $MAE\_{HR}$ ↓ | rCLIP Score ↑       |
> |--------------------|--------------------|------------------------|---------------------|
> | **SE-Diff (ours)** (DDPM) | 0.0714 ± 0.0010    | **8.43 ± 0.42**        | **0.9470 ± 0.0029** |
> | DDIM               | 0.1219 ± 0.0010    | 9.05 ± 0.44            | 0.7783 ± 0.0072     |
> | Second Order       | 0.0877 ± 0.0011    | 13.59 ± 1.05           | 0.8671 ± 0.0040     |
> | SDE                | **0.0693 ± 0.0106**| 19.21 ± 1.27           | 0.9213 ± 0.0028     |
>
> **Effects of noise schedule.**
>
> | Noise Schedule | NRMSE ↓            | $MAE\_{HR}$ ↓ | rCLIP Score ↑       |
> |----------------|--------------------|------------------------|---------------------|
> | Linear         | **0.0714 ± 0.0010**| **8.43 ± 0.42**        | **0.9470 ± 0.0029** |
> | Cosine         | 0.0728 ± 0.0032    | 8.90 ± 0.78            | 0.9446 ± 0.0131     |
>
> We have added the above experiments and discussion in Table 6&7&8 in Appendix I.

---

### Official Review · Reviewer_R6Yk · 2025-10-27

**Soundness:** 3
**Presentation:** 4
**Contribution:** 2
**Rating:** 2
**Confidence:** 4

**Summary:**

This paper proposed SE-Diff which generates 10-second, 12-lead ECGs by running conditional 1D diffusion in a VAE latent space augmented with an ECG ODE simulator (physics priors) and EHR-based experience (RAG).
Text, metadata, and retrieved reports are injected via cross-attention and FiLM, and Euler/inter-lead constraints are applied on the beat-decoder output to enforce physiological consistency.
On MIMIC-IV-ECG, SE-Diff outperforms DiffuSETS across signal-level, alignment, and distribution-level

**Strengths:**

The dual-guidance design (physics + experience) cleanly integrates simulator regularization and EHR-RAG conditioning to jointly improve morphology preservation and text-to-ECG alignment within a well-structured latent-diffusion pipeline.

**Weaknesses:**

- Baseline coverage is narrow because comparisons center primarily on DiffuSETS instead of including strong adjacent baselines such as unconditional diffusion, flow-matching, GAN/SSM, or ODE-aware variants (e.g.,  https://www.sciencedirect.com/science/article/abs/pii/S1746809425010559
, https://proceedings.neurips.cc/paper_files/paper/2024/file/9988f2c8e07c1f98af7ba9ca31ccae0b-Paper-Conference.pdf
, https://arxiv.org/html/2409.17833v1
).

- Case analyses would be stronger with quantitative morphology and interval metrics per case, such as PR, QRSd, QT/QTc, ST shift/slope, and P/T amplitude/width.

- Evidence for domain generalization is limited because results concentrate on MIMIC-IV-ECG, so small external cohort or device tests reporting rFID, HR-MAE, and interval MAE would better support robustness.

**Questions:**

- Can you add unconditional/flow-matching/GAN·SSM/ODE-aware baselines under the same length/leads/conditioning/steps to demonstrate generality?

- Can you report per-case PR, QRSd, QT/QTc, ST shift/slope, and P/T features to quantify morphology and interval fidelity?

- Can you show external-cohort/device results with rFID, HR-MAE, and interval MAE to evidence domain generalization and robustness?

---

> ### Author Response · Authors · 2025-11-24
>
> > **W1: Baseline coverage is narrow because comparisons center primarily on DiffuSETS instead of including strong adjacent baselines such as unconditional diffusion, flow-matching, GAN/SSM, or ODE-aware variants.**
> >
> > **Q1: Can you add unconditional/flow-matching/GAN·SSM/ODE-aware baselines under the same length/leads/conditioning/steps to demonstrate generality?**
>
> We thank the reviewer for the helpful suggestion on broadening baseline coverage. In the revision, we have added several strong adjacent baselines for 10 s, 12-lead ECG generation using publicly available code: (i) a GAN-based model (WGAN) for ECG augmentation [Adib et al., BIBM 2022], (ii) a diffusion model with a structured state-space backbone (SSSD) [López Alcaraz & Strodthoff, CBM 2023], and (iii) a beat-level ECG diffusion model with morphology-aware reconstruction (BeatDiff) [Bedin et al., NeurIPS 2024]. We re-implemented and trained these methods under our setting and reported results in the updated tables.
>
> **ECG generation performance on MIMIC-IV-ECG dataset.**
>
> | Model             | MAE ↓              | NRMSE ↓           | $MAE\_{HR}$ ↓          | rCLIP Score ↑       | rFID Score ↑        |
> |-------------------|--------------------|-------------------|--------------------|---------------------|---------------------|
> | SSDM              | 0.4337 ± 0.0300    | 0.2027 ± 0.0441   | 27.37 ± 14.84      | 0.7213 ± 0.0402     | 0.9096 ± 0.0398     |
> | WGAN              | 0.1896 ± 0.0605    | 0.1301 ± 0.0316   | 31.54 ± 2.15       | 0.5688 ± 0.0347     | 0.5497 ± 0.0192     |
> | BeatDiff          | 0.7464 ± 0.0070    | 0.4756 ± 0.0117   | 27.74 ± 1.49       | 0.5167 ± 0.0180     | 0.8612 ± 0.0039     |
> | DiffuSETS         | 0.1092 ± 0.0022    | 0.0851 ± 0.0012   | 13.29 ± 1.13       | 0.9309 ± 0.0036     | 0.9209 ± 0.0038     |
> | **SE-Diff (ours)** | **0.0923 ± 0.0021** | **0.0714 ± 0.0010** | **8.43 ± 0.42**     | **0.9470 ± 0.0029** | **0.9509 ± 0.0035** |
> | w/o Exp           | 0.0926 ± 0.0022    | 0.0730 ± 0.0008   | 15.06 ± 0.34       | 0.9099 ± 0.0026     | 0.9032 ± 0.0062     |
> | w/o InterLead     | 0.0934 ± 0.0023    | 0.0733 ± 0.0006   | 19.21 ± 1.27       | 0.9216 ± 0.0029     | 0.9128 ± 0.0052     |
> | w/o Sim           | 0.0965 ± 0.0024    | 0.0768 ± 0.0014   | 14.28 ± 1.35       | 0.9303 ± 0.0041     | 0.9138 ± 0.0056     |
>
> In addition, we use the same set of generators (SSDM, WGAN, BeatDiff, DiffuSETS, SE-Diff) as synthetic minority over-samplers in the downstream ECG classification experiments under severe class imbalance (gender classification and rare SVT detection). As shown in the table below, all generative baselines improve over the unbalanced real-only classifier, while SE-Diff yields the largest gains and consistently closes the gap toward the fully balanced real-only upper bound.
>
> **Downstream ECG classification under severe class imbalance.**
>
> | Model      | F1 (Male=10% Female) ↑ | Acc (Male=10% Female) ↑ | AUC (Male=10% Female) ↑ | F1 (SVT=10% Sinus) ↑ | Acc (SVT=10% Sinus) ↑ | AUC (SVT=10% Sinus) ↑ |
> |-----------|-------------------------|--------------------------|--------------------------|----------------------|------------------------|------------------------|
> | SSDM      | 42 ± 0                  | 54 ± 1                   | 49 ± 2                   | 56 ± 1               | 63 ± 2                 | 81 ± 2                 |
> | WGAN      | 42 ± 0                  | 54 ± 1                   | 49 ± 2                   | 57 ± 2               | 63 ± 2                 | 82 ± 2                 |
> | BeatDiff  | 44 ± 2                  | 55 ± 2                   | 51 ± 2                   | 60 ± 2               | 67 ± 2                 | 84 ± 3                 |
> | DiffuSETS | 44 ± 3                  | 54 ± 2                   | 54 ± 1                   | 70 ± 1               | 68 ± 1                 | 84 ± 2                 |
> | SE-Diff (ours) | **58 ± 1**         | **58 ± 1**               | **58 ± 2**               | **72 ± 2**           | **71 ± 0**             | **85 ± 2**             |
> | Unbalanced     | 42 ± 0             | 54 ± 1                   | 46 ± 2                   | 56 ± 1               | 62 ± 0                 | 80 ± 1                 |
> | Balanced       | 62 ± 0             | 62 ± 0                   | 62 ± 1                   | 79 ± 1               | 80 ± 2                 | 93 ± 1                 |

---

> > ### Author Response · Authors · 2025-11-24
> >
> > > **W2: Case analyses would be stronger with quantitative morphology and interval metrics per case, such as PR, QRSd, QT/QTc, ST shift/slope, and P/T amplitude/width.**
> > >
> > > **Q2: Can you report per-case PR, QRSd, QT/QTc, ST shift/slope, and P/T features to quantify morphology and interval fidelity?**
> >
> >
> > We thank the reviewer for this helpful suggestion. In the revised manuscript, we have added quantitative morphology and interval metrics to strengthen the case analyses. Specifically, in addition to the global ECG generation metrics (MAE, NRMSE, HR MAE, rCLIP, rFID) reported in Table 2 (reproduced below), we now report Beat-level morphology and interval fidelity (record-level median MAE for PR, QRSd, QT, QTcF, ST@J+60, P duration, and T duration) on MIMIC-IV-ECG in a new Table 3. As shown, SE-Diff achieves consistently lower errors on these clinically meaningful features than all baselines (SSDM, WGAN, BeatDiff, DiffuSETS), and the ablations (w/o Sim, w/o InterLead, w/o Exp) further confirm the contribution of the proposed modules to morphology and interval fidelity.
> >
> >
> > **Beat-level morphology & interval fidelity (MAE) on MIMIC-IV-ECG (record-level medians; lower is better).**
> >
> > | Model        | PR ↓         | QRSd ↓       | QT ↓         | QTcF ↓       | ST@J+60 ↓   | P dur ↓      | T dur ↓       |
> > |--------------|--------------|--------------|--------------|--------------|-------------|--------------|---------------|
> > | SSDM         | 7.70 ± 8     | 32 ± 1       | 25.11 ± 11   | 23.25 ± 11   | 0.16 ± 0    | 5.30 ± 3     | 23.71 ± 9     |
> > | WGAN         | 18.01 ± 9    | 29.64 ± 9    | 21.02 ± 5    | 23.83 ± 2    | 0.13 ± 0    | 6.01 ± 0     | 24.01 ± 4     |
> > | BeatDiff     | 10.34 ± 5    | 18.23 ± 6    | 14.23 ± 6    | 9.43 ± 5     | 0.17 ± 0    | 7.00 ± 1     | 30.02 ± 3     |
> > | DiffuSETS    | 14.81 ± 6    | 11.71 ± 7    | 8.20 ± 3     | 9.71 ± 4     | 0.04 ± 0    | 5.60 ± 1     | 8.71 ± 7      |
> > | **SE-Diff** | **7.30 ± 3** | **10.71 ± 4**| **4.50 ± 2** | **7.88 ± 2** | **0.03 ± 0**| **2.50 ± 0** | **6.80 ± 3**  |
> > | w/o InterLead| 11.11 ± 3    | 15.01 ± 5    | 11.91 ± 5    | 15.84 ± 7    | 0.03 ± 0    | 4.00 ± 1     | 8.10 ± 4      |
> > | w/o Exp      | 9.81 ± 9     | 14.11 ± 13   | 8.20 ± 3     | 12.61 ± 3    | 0.04 ± 0    | 4.00 ± 1     | 6.90 ± 3      |
> > | w/o Sim      | 12.11 ± 7    | 13.21 ± 6    | 8.40 ± 4     | 8.89 ± 7     | 0.04 ± 0    | 5.10 ± 1     | 13.11 ± 1     |

---

> ### Author Response · Authors · 2025-11-24
>
> > **W3: Evidence for domain generalization is limited because results concentrate on MIMIC-IV-ECG, so small external cohort or device tests reporting rFID, HR-MAE, and interval MAE would better support robustness.**
> >
> > **Q3: Can you show external-cohort/device results with rFID, HR-MAE, and interval MAE to evidence domain generalization and robustness?**
>
> We thank the reviewer for highlighting the need for stronger evidence of domain generalization. In the revised manuscript, we have added experiments on a second real-world cohort, the PTB-XL dataset, which differs from MIMIC-IV-ECG in patient population, labeling scheme, and acquisition characteristics. For PTB-XL, we report the same set of metrics as on MIMIC-IV-ECG, including rFID, HR-MAE, and Beat-level morphology/interval MAE (PR, QRSd, QT/QTcF, ST@J+60, P/T durations). Across all of these metrics, SE-Diff consistently outperforms recent GAN-, SSM-, and diffusion-based baselines.
>
>
> **ECG generation performance on PTB-XL dataset.**
>
> | Model             | MAE ↓              | NRMSE ↓           | $MAE\_{HR}$ ↓          | rCLIP Score ↑       | rFID Score ↑        |
> |-------------------|--------------------|-------------------|--------------------|---------------------|---------------------|
> | SSDM              | 0.6103 ± 0.0204    | 0.3818 ± 0.0670   | 15.22 ± 11.51      | 0.8618 ± 0.0599     | 0.7168 ± 0.0355     |
> | WGAN              | 0.2458 ± 0.0653    | 0.1197 ± 0.0313   | 13.82 ± 18.69      | 0.5880 ± 0.0000     | 0.5377 ± 0.0232     |
> | BeatDiff          | 0.9888 ± 0.0059    | 0.4731 ± 0.0104   | 13.86 ± 0.78       | 0.8799 ± 0.0022     | 0.8503 ± 0.0035     |
> | DiffuSETS         | 0.1281 ± 0.0030    | 0.0797 ± 0.0011   | 17.88 ± 0.72       | 0.8690 ± 0.0011     | 0.8456 ± 0.0035     |
> | **SE-Diff (ours)** | **0.1076 ± 0.0033** | **0.0630 ± 0.0006** | **8.24 ± 0.43**     | **0.8901 ± 0.0060** | **0.8583 ± 0.0056** |
> | w/o Sim           | 0.1138 ± 0.0032    | 0.0680 ± 0.0007   | 14.72 ± 0.90       | 0.8896 ± 0.0010     | 0.8004 ± 0.0061     |
> | w/o InterLead     | 0.1084 ± 0.0034    | 0.0640 ± 0.0007   | 12.02 ± 0.78       | 0.7484 ± 0.0076     | 0.8568 ± 0.0051     |
>
> **Beat-level morphology & interval fidelity (MAE) on PTB-XL (record-level medians; lower is better).**
>
> | Model        | PR ↓         | QRSd ↓       | QT ↓         | QTcF ↓       | ST@J+60 ↓   | P dur ↓      | T dur ↓       |
> |--------------|--------------|--------------|--------------|--------------|-------------|--------------|---------------|
> | SSDM         | 13.71 ± 8    | 13.61 ± 8    | 30.22 ± 9    | 27.69 ± 14   | 0.33 ± 0    | 12.71 ± 5    | 19.11 ± 10    |
> | WGAN         | 14.01 ± 7    | 17.45 ± 13   | 12.01 ± 10   | 12.97 ± 10   | 0.18 ± 0    | 7.33 ± 3     | 25.68 ± 13    |
> | BeatDiff     | 13.41 ± 4    | 17.40 ± 6    | 9.81 ± 3     | 8.94 ± 2     | 0.73 ± 1    | 4.10 ± 1     | 18.81 ± 6      |
> | DiffuSETS    | 7.70 ± 2     | 12.61 ± 7    | 11.51 ± 4    | 13.75 ± 5    | 0.10 ± 0    | 9.11 ± 6     | 13.20 ± 5     |
> | **SE-Diff** | **3.90 ± 1** | **10.01 ± 2**| **5.20 ± 1** | **8.68 ± 2** | **0.07 ± 0**| **3.00 ± 1** | **9.31 ± 8**  |
> | w/o InterLead| 7.00 ± 1     | 10.31 ± 5    | 8.40 ± 4     | 10.91 ± 3    | 0.07 ± 0    | 4.40 ± 1     | 13.61 ± 8     |
> | w/o Sim      | 9.21 ± 4     | 12.91 ± 5    | 12.51 ± 4    | 14.20 ± 2    | 0.07 ± 0    | 4.50 ± 1     | 12.31 ± 3     |
>
>
> We have added the above experiments and discussion in Table 1&2&3 in Section 4.1&4.2.
>
>
> Reference:
>
> 1. WGAN: Adib, E., Afghah, F., and Prevost, J. J. (2022). Arrhythmia classification using cgan-augmented
> ecg signals*. 2022 IEEE International Conference on Bioinformatics and Biomedicine (BIBM),
> pages 1865–1872.
> 2. SSSD: Alcaraz, J. M. L. and Strodthoff, N. (2023). Diffusion-based conditional ecg generation with
> structured state space models. Computers in Biology and Medicine, 163:107115
> 3. Bedin, Lisa, et al. "Leveraging an ECG beat diffusion model for morphological reconstruction from indirect signals." Advances in Neural Information Processing Systems 37 (2024): 84409-84446.

---

### Official Review · Reviewer_f2NA · 2025-10-30

**Soundness:** 3
**Presentation:** 2
**Contribution:** 3
**Rating:** 6
**Confidence:** 4

**Summary:**

This article presents SE-Diff, a diffusion model for ECG generation enhanced by physiological simulator and retrieved clinical records. Experiments show that SE-Diff prevails over current ECG generation baseline and is competent for enhancing downstream ECG screening model.

**Strengths:**

1. The method of combining ODE-based physiological simulator with diffusion-based generative learning is novel in ECG synthesis and provides interpretability and physiological plausibility.
2. Comprehensive evaluation across fidelity, semantic alignment, and downstream classification demonstrates consistent improvement over baselines.
3. Visualization of model generated ECG and single-cycle waveform from Beat Decoder provide clear evidence to model efficacy and the validity of proposed method.

**Weaknesses:**

1. Description of “experience retrieved-augmented condition” is too brief. In the main paper, the condition comprises $(t, m, r)$ while the appendix only refers the concatenation of text token and meta token, which may confound the readers who are not familiar with conditional ECG generation. Moreover, what is value of top-k’s k in practical?
2. The quality of the “experience-augmented” text conditioning depends heavily on LLM summarization, but neither the type of LLM nor the output case are shown in paper.
3. In the ECG generation results section, a proper discussion to the model performance is absent, which may critically hinder a further understanding to the benefit of designed module.
4. The inference efficiency of model is not reported.
5. The entire work is based on MIMIC ECG dataset, the generalization ability of proposed method is not discussed.

**Questions:**

See Weakness.

---

> ### Author Response · Authors · 2025-11-24
>
> > **W1: Description of “experience retrieved-augmented condition” is too brief. In the main paper, the condition comprises $(t, m, r)$ while the appendix only refers the concatenation of text token and meta token, which may confound the readers who are not familiar with conditional ECG generation. Moreover, what is value of top-k’s k in practical?**
>
> We thank the reviewer for pointing out that our description of the experience retrieval–augmented conditioning was too brief. In our model, the final conditioning input to the denoiser is $c = (t,m,r)$, where $t$ is the original diagnosis/report text, $m$ is a short natural-language template of metadata (age, sex, optionally heart rate), and $r$ is an LLM-generated “experience” report based on the current case and retrieved similar admissions.
>
> To construct $r$, we first retrieve the top-$k$ most similar admissions according to the tri-view similarity over diagnoses, medications, and procedures, then pass their diagnostic profiles together with $(t,m)$ into the LLM using the prompt shown in Fig. 5 in Appendix to obtain a concise, physiologically grounded report.
>
> In all experiments we set $k = 3$, which we found to balance informativeness and prompt length: $k = 1$ can be too case-specific, while larger $k$ values mainly increase prompt size without clear gains.
>
> We have added a more detailed description of the experience retrieval–augmented conditioning to Section 3.4 and clarified the choice of $k$ in Appendix A.4 to make this pipeline clearer to readers.
>
>
> > **W2: The quality of the “experience-augmented” text conditioning depends heavily on LLM summarization, but neither the type of LLM nor the output case are shown in paper.**
>
> We thank the reviewer for highlighting that the description of the LLM used for experience-augmented conditioning was incomplete. In this work, we use an OpenAI model (GPT-4.1-mini, accessed via the OpenAI API), chosen to balance summarization quality and computational cost. In addition, we have added a complete illustrative example of the experience retrieval–augmented conditioning process:
>
> ***Illustrative example of experience retrieval–augmented conditioning.***
>
> - ****Original machine-generated diagnosis and metadata.****
>   - Original machine-generated diagnosis (most important first): *Sinus rhythm*
>   - Heart rate (bpm): 83
>   - Age (years): 91
>   - Gender: F
>
> - ****Top-$k$ retrieved similar admissions (diagnosis view, $k=3$).****
>   Retrieved similar patients' diagnoses:
>   1. *Sinus rhythm*
>   2. *Sinus rhythm*
>   3. *Sinus rhythm with borderline 1st degree A–V block*
>
> - ****LLM-generated experience report.****
> The ECG demonstrates a sinus rhythm at a rate of 83 beats per minute in a 91-year-old female. The rhythm is regular with normal P wave morphology preceding each QRS complex, consistent with sinus node origin. There is no evidence of atrioventricular conduction delay or block; PR interval is within normal limits. QRS complexes are narrow, indicating normal ventricular conduction. No arrhythmias or ectopic beats are noted. Overall, the tracing reflects a stable sinus rhythm appropriate for the patient's age without signs of sinus tachycardia or conduction abnormalities.
>
>
> We have added the above example in Appendix L.
>
> > **W3: In the ECG generation results section, a proper discussion to the model performance is absent, which may critically hinder a further understanding to the benefit of designed module.**
>
> We thank the reviewer for pointing out the lack of discussion on the ECG generation results. In the revised manuscript, we have added a dedicated analysis of the ECG generation experiment.
>
>
> > **W4: The inference efficiency of model is not reported.**
>
> We thank the reviewer for raising the question about inference efficiency. We have added an inference latency analysis to the appendix, reporting the time required to generate one 10 s, 12-lead ECG under different batch sizes with 1000 diffusion sampling steps; the results are shown in the table below.
>
> | **Batch size** | **Time / batch (s)** | **Time / sample (s)** |
> |---------------:|---------------------:|----------------------:|
> | 32             | 13.55                | 0.425                 |
> | 128            | 13.65                | 0.105                 |
> | 256            | 13.40                | 0.050                 |
> | 512            | 15.05                | 0.0295                |
> | 1024           | 24.90                | 0.0245                |
> | 2048           | 45.00                | 0.0220                |
> | 4096           | 84.75                | 0.0205                |
>
> We have added the above experiments and discussion in Table 5 in Appendix H.

---

> ### Author Response · Authors · 2025-11-24
>
> > **W5: The entire work is based on MIMIC ECG dataset, the generalization ability of proposed method is not discussed.**
>
> We thank the reviewer for raising the question about generalization beyond MIMIC-IV-ECG. In the revised manuscript, we have added experiments on the PTB-XL dataset, including both ECG generation performance and morphology & interval fidelity (MAE), and we also compare against additional recent baselines (SSDM, WGAN, BeatDiff). As shown below, SE-Diff consistently outperforms all baselines on PTB-XL across signal-level metrics (MAE, NRMSE, HR MAE, rCLIP, rFID) and on clinically meaningful morphology/interval metrics, demonstrating that the proposed simulator-informed and experience-augmented conditioning generalize well beyond a single dataset.
>
> **ECG generation performance on PTB-XL dataset.**
>
> | Model             | MAE ↓              | NRMSE ↓           | $MAE\_{HR}$ ↓          | rCLIP Score ↑       | rFID Score ↑        |
> |-------------------|--------------------|-------------------|--------------------|---------------------|---------------------|
> | SSDM              | 0.6103 ± 0.0204    | 0.3818 ± 0.0670   | 15.22 ± 11.51      | 0.8618 ± 0.0599     | 0.7168 ± 0.0355     |
> | WGAN              | 0.2458 ± 0.0653    | 0.1197 ± 0.0313   | 13.82 ± 18.69      | 0.5880 ± 0.0000     | 0.5377 ± 0.0232     |
> | BeatDiff          | 0.9888 ± 0.0059    | 0.4731 ± 0.0104   | 13.86 ± 0.78       | 0.8799 ± 0.0022     | 0.8503 ± 0.0035     |
> | DiffuSETS         | 0.1281 ± 0.0030    | 0.0797 ± 0.0011   | 17.88 ± 0.72       | 0.8690 ± 0.0011     | 0.8456 ± 0.0035     |
> | **SE-Diff (ours)** | **0.1076 ± 0.0033** | **0.0630 ± 0.0006** | **8.24 ± 0.43**     | **0.8901 ± 0.0060** | **0.8583 ± 0.0056** |
> | w/o Sim           | 0.1138 ± 0.0032    | 0.0680 ± 0.0007   | 14.72 ± 0.90       | 0.8896 ± 0.0010     | 0.8004 ± 0.0061     |
> | w/o InterLead     | 0.1084 ± 0.0034    | 0.0640 ± 0.0007   | 12.02 ± 0.78       | 0.7484 ± 0.0076     | 0.8568 ± 0.0051     |
>
> **Beat-level morphology & interval fidelity (MAE) on PTB-XL (record-level medians; lower is better).**
>
> | Model        | PR ↓         | QRSd ↓       | QT ↓         | QTcF ↓       | ST@J+60 ↓   | P dur ↓      | T dur ↓       |
> |--------------|--------------|--------------|--------------|--------------|-------------|--------------|---------------|
> | SSDM         | 13.71 ± 8    | 13.61 ± 8    | 30.22 ± 9    | 27.69 ± 14   | 0.33 ± 0    | 12.71 ± 5    | 19.11 ± 10    |
> | WGAN         | 14.01 ± 7    | 17.45 ± 13   | 12.01 ± 10   | 12.97 ± 10   | 0.18 ± 0    | 7.33 ± 3     | 25.68 ± 13    |
> | BeatDiff     | 13.41 ± 4    | 17.40 ± 6    | 9.81 ± 3     | 8.94 ± 2     | 0.73 ± 1    | 4.10 ± 1     | 18.81 ± 6      |
> | DiffuSETS    | 7.70 ± 2     | 12.61 ± 7    | 11.51 ± 4    | 13.75 ± 5    | 0.10 ± 0    | 9.11 ± 6     | 13.20 ± 5     |
> | **SE-Diff** | **3.90 ± 1** | **10.01 ± 2**| **5.20 ± 1** | **8.68 ± 2** | **0.07 ± 0**| **3.00 ± 1** | **9.31 ± 8**  |
> | w/o InterLead| 7.00 ± 1     | 10.31 ± 5    | 8.40 ± 4     | 10.91 ± 3    | 0.07 ± 0    | 4.40 ± 1     | 13.61 ± 8     |
> | w/o Sim      | 9.21 ± 4     | 12.91 ± 5    | 12.51 ± 4    | 14.20 ± 2    | 0.07 ± 0    | 4.50 ± 1     | 12.31 ± 3     |
>
>
> We have added the above experiments and discussion in Table 1&2 in Section 4.1.

---

### Author Response · Authors · 2025-11-24
**General Response to All Reviewers:**

We thank all the reviewers for their insightful questions and constructive suggestions! We are glad that the reviewers found our paper “novel in ECG synthesis”, “provides interpretability and physiological plausibility”, “comprehensive evaluation across fidelity, semantic alignment, and downstream classification”, “dual-guidance design (physics + experience)”, “dual-channel injection of mechanism priors and empirical knowledge”, “solid quantitative results”, “downstream value demonstration”, “timely and relevant topic”, and “clear, carefully engineered writing”.

We have addressed each reviewer's concerns individually in our responses to their reviews. Below is a summary of the major revisions:

1. **Broader baselines:** We add strong adjacent ECG-generation baselines (WGAN, SSDM, BeatDiff, and DiffuSETS) under the same 10 s, 12-lead setting and show that SE-Diff consistently outperforms them.

2. **External validation on PTB-XL:** We conduct external validation on PTB-XL, reporting the same metrics as on MIMIC-IV-ECG and demonstrating that SE-Diff maintains its advantages across datasets with different populations, labels, and acquisition characteristics.

3. **Richer evaluation:** We augment the evaluation with Beat-level morphology and interval fidelity metrics (PR, QRSd, QT/QTcF, ST@J+60, P and T durations).

4. **Inference efficiency analysis:** We report inference time for generating 10 s, 12-lead ECGs across different batch sizes and sampling settings, quantifying the computational cost and scalability of SE-Diff.

5. **Clarified experience-guided conditioning:** We more clearly describe the experience retrieval–augmented conditioning pipeline c = (t, m, r).

6. **Sensitivity analyses on retrieval hyperparameters λ1, λ2, λ3 and top-k:** We perform sensitivity analyses over the tri-view weights λ1, λ2, λ3 and the number of retrieved admissions (for example, k in {2, 3, 4}), showing that SE-Diff is robust to these choices and that the reported setting λ1 = λ2 = λ3 = 1, k = 3 offers a good performance–cost trade-off.

7. **rCLIP/rFID under multiple ECG/text encoder configurations:** We evaluate SE-Diff and DiffuSETS under several ECG/text encoder variants and show that SE-Diff consistently outperforms DiffuSETS in rCLIP/rFID as well as MAE and HR-MAE, indicating that our conclusions are not tied to a particular encoder choice.

8. **Ethics and privacy discussion:** We expand the ethics/privacy section to clarify patient-level splitting, restriction of retrieval to the training cohort, use of de-identified codes and LLM-generated summaries (rather than raw free text), and how external-cohort performance supports utility without training–test leakage or memorization.

All revisions are included in the revised paper highlighted in orange. We appreciate the reviewers’ thoughtful feedback again, which helped strengthen the paper. Should you have any further questions, please do not hesitate to let us know. We are fully prepared to address any additional concerns that may arise.

---

### Meta-Review · Area_Chair_4gRS · 2026-01-09

**Summary:**

This paper proposes SE-Diff, a simulator- and experience-enhanced diffusion framework for comprehensive ECG generation from clinical context. By integrating a lightweight physiological ODE-based ECG simulator into the diffusion process and augmenting conditioning with experience retrieved from EHRs and distilled via an LLM, the method injects both mechanistic priors and empirical clinical knowledge. Reviewers agreed that the problem is timely and important, and that the dual-guidance design is novel and well motivated. Extensive experiments demonstrate consistent improvements over strong baselines in signal fidelity, text–ECG alignment, and downstream ECG classification, supporting the practical value of the approach. The AC recommends acceptance.

**Reviewer Concerns:**

Reviewers initially raised concerns regarding baseline coverage, clarity of the experience-augmented conditioning, evaluation scope, reproducibility details, and evidence for domain generalization. These concerns were thoroughly addressed in the rebuttal and revised manuscript. The authors added multiple strong adjacent baselines (GAN-, SSM-, and diffusion-based), expanded evaluation with clinically meaningful beat-level morphology and interval metrics, reported inference efficiency, and conducted external validation on the PTB-XL dataset. Reproducibility details (hyperparameters, prompts, sensitivity analyses) and ethics/privacy considerations were substantially clarified. While some limitations remain (e.g., simulator constraints for precordial leads and reliance on specific encoders), reviewers acknowledged that the revisions resolved the major technical issues and significantly strengthened the paper.

**Reviewer Scores:**

Reviewer scores ranged from marginal to clear accept. Although one reviewer initially recommended rejection (score 2), their key concerns were explicitly addressed in the revision.

---

### Decision · Program_Chairs · 2026-01-26

Accept (Poster)